# u-HuBERT: Unified Mixed-Modal Speech Pretraining And Zero-Shot Transfer to Unlabeled Modality

**Wei-Ning Hsu**
Meta AI
wnhsu@meta.com

**Bowen Shi**
Meta AI
bshi@meta.com

## Abstract

While audio-visual speech models can yield superior performance and robustness compared to audio-only models, their development and adoption are hindered by the lack of labeled and unlabeled audio-visual data and the cost to deploy one model per modality. In this paper, we present u-HuBERT, a self-supervised pre-training framework that can leverage both multimodal and unimodal speech with a unified masked cluster prediction objective. By utilizing modality dropout during pre-training, we demonstrate that a single fine-tuned model can achieve performance on par or better than the state-of-the-art modality-specific models. Moreover, our model fine-tuned only on audio can perform well with audio-visual and visual speech input, achieving zero-shot modality generalization for multiple speech processing tasks. In particular, our single model yields 1.2%/1.4%/27.2% speech recognition word error rate on LRS3 with audio-visual/audio/visual input.[1]

## 1 Introduction

Speech processing has been moving from developing domain-specific models toward building a general model for all domains. For example, early studies presented models dedicated for robust speech recognition [Kingsbury et al., 1998] and distant speech recognition [Feng et al., 2014, Kim et al., 2017] by taking into account acoustic properties for each domain. In contrast, recent work has built general domain models by pooling data from different domains and even different languages, including labeled ones for supervised learning [Chan et al., 2021, Likhomanenko et al., 2020, Pratap et al., 2020] and unlabeled ones for self-supervised learning [Kawakami et al., 2020, Hsu et al., 2021b, Conneau et al., 2020]. These studies show that in addition to removing the need of deploying and maintaining multiple models, a single general domain model can in fact achieve performance on par or superior to domain-specific models, especially for domains with less data, and improve generalization performance to unseen domains.

In contrast, there are few attempts and even fewer successful ones on developing a versatile model that can process many speech modalities [Makino et al., 2019, Sari et al., 2021]. In general, researchers develop models with different input speech modalities, such as visual speech [Chung and Zisserman, 2016], audio-visual speech [Xu et al., 2020], multi-channel speech [Haeb-Umbach et al., 2021], 3D talking faces [Richard et al., 2021], Electromyographic (EMG) muscle signals [Diener and Schultz, 2018], ultrasonic oral cavity scan [Hueber et al., 2010], or ultrasonic microphone [Livescu et al., 2009] in isolation, despite the fact that many of them are used to solve the same task. Moreover, while additional modalities, such as visual speech, can often boost the performance and robustness on top of audio speech, models taking such input are less studied and deployed in real world application. This is mainly due to the lack of both labeled and unlabeled data for these alternative modalities, hindering the development of high-performing self-supervised or supervised modal-specific models that are

---

[1] Codes and models are available at `https://github.com/facebookresearch/av_hubert`

36th Conference on Neural Information Processing Systems (NeurIPS 2022).

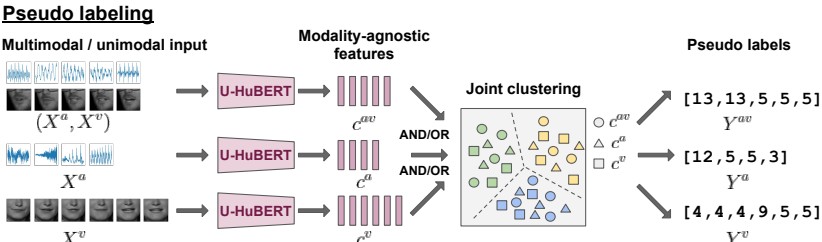

Figure 1: u-HuBERT learns modality-agnostic features that are clustered to produce pseudo labels with a shared codebook for different speech modalities. See Section 3.2 for more information.

trained exclusively on one type of input. We argue that to tackle the data scarcity issue, it essential to utilize not only the speech from the target modality, but also speech in other modalities to build a general speech model for many modalities. By doing so, more data can be used to learn components shared across different modalities, which is expected to bring bigger gains to low-resource modalities.

Audio-visual HuBERT (AV-HuBERT) [Shi et al., 2022a] is a recent work progressing toward creating a versatile speech model: by pre-training on audio-visual speech, it demonstrates strong performance for not only audio-visual downstream tasks, but also audio-only and video-only tasks. However, it only explores pre-training exclusively on audio-visual speech, and fine-tuning and testing on matching modalities. In this paper, we generalize AV-HuBERT and present unified hidden unit BERT (u-HuBERT), which leverages unlabeled speech data of many different modalities for pre-training, including both unimodal and multimodal speech. The model is trained with a unified masked cluster prediction objective for both unimodal and multimodal speech data. The pre-trained u-HuBERT can be fine-tuned with either multimodal or unimodal data, and we demonstrate that in both cases the resulting model can be used for many speech modalities. Following AV-HuBERT, we apply modality dropout [Neverova et al., 2014] to multimodal data to simulate different combinations of input modalities. We present extensive analysis in this paper to show that modality dropout is crucial for learning modality-agnostic frame-level representation, which is the key to successful mixed modality pre-training and achieving good performance when fine-tuning and testing modalities mismatch. We verify the effectiveness on two popular speech processing tasks: speech recognition and speech translation. To the best of our knowledge, this is the first single model for many speech modalities that yields performance on par with the state-of-the-art modality-specific models and is capable of zero-shot modality generalization.

## 2   Related Work

**Comparison to [Shi et al., 2022a]**   u-HuBERT generalizes AV-HuBERT to utilize both multimodal and unimodal speech for pre-training by mapping various input to a modality-agnostic feature space and creating a shared target space for masked prediction. In contrast, because AV-HuBERT is pre-trained exclusively on audio-visual speech, it always generates targets using features of the same modality (audio in the first iteration and audio-visual for the subsequent iterations).[2] Therefore, learning modality-agnostic features to produce shared target space is not essential for AV-HuBERT.

Moreover, this paper presents a novel evaluation scenario: zero-shot transfer to unlabeled modalities, which is of great practical importance because while labeled data may not always be available in the target modality, it will most likely be available in some other modalities. We demonstrate in this paper that modality dropout is key to achieving good zero-shot transfer performance.

**Joint unimodal and multimodal pre-training**   Recent cross-modal contrastive approaches such as CLIP [Radford et al., 2021] and ALIGN [Jia et al., 2021] have shown impressive results yet they can only be trained with multimodal data and require a large-scale dataset, which can be hard to obtain for many speech modalities (e.g., paired EMG-audio speech). FLAVA [Singh et al., 2021] and SLAM [Bapna et al., 2021] are two frameworks that utilize both unimodal and multimodal data

---

[2]AV-HuBERT predicts the same audio-visual cluster targets $C^{av}$ regardless of whether modality dropout is activated. See the top half of Figure 2.

for image-text and speech-text pre-training, respectively. There are three key differences between u-HuBERT and these two models. First, for multimodal speech, different modalities such as audio and visual speech are temporally aligned. Instead of concatenating the feature sequences from unimodal encoders, we fuse the features frame-by-frame before passing it to the shared encoder. Second, while all three models utilize a masked token prediction objective, u-HuBERT considers a shared target token space for all unimodal and multimodal data, obtained by clustering modality-agnostic features extracted from the u-HuBERT model itself. In contrast, SLAM uses w2v-BERT [Chung et al., 2021] tokens for speech and SentencePiece [Schuster and Nakajima, 2012] for text. FLAVA on other hand uses a separately trained dVAE tokenizer for image and the same as SLAM for text, and it predicts tokens for unimodal and multimodal data at the output of different modules, differing from u-HuBERT. In addition, both FLAVA and SLAM pre-train with a number of auxiliary objectives, while u-HuBERT only uses one, which simplifies hyperparameter selection. Third, different modalities of speech are often used to solve the same task. Our evaluation focuses on building a single model that can solve the same task with different modalities despite not having labeled data for each modality, which is not considered by SLAM or FLAVA.

**Single model for many input modalities** Makino et al. [2019] explored building a single model for supervised audio-visual/audio/visual speech recognition. The studies show significant degradation comparing unified models with modality-specific ones. Furthermore, these models are single-task supervised and hence not capable of zero-shot modality transfer when trained on unimodal data. Omnivore [Girdhar et al., 2022] is a single supervised model for many vision modalities, including image (RGB), 3D image (RGBD), and video (RGB sequence), which also considers a model architecture with modality-specific encoders (RGB and depth) followed by a shared Swin Transformer backend [Liu et al., 2021]. Both Omnivore and u-HuBERT demonstrates cross-modal generalization, but Omnivore achieves it through supervised learning with different tasks, and u-HuBERT achieves it with unified self-supervised learning. Finally, if we considered different languages as different modalities, our work is also related to mBERT [Devlin et al., 2019], mBART [Liu et al., 2020], TLM [Yao et al., 2021], and XLM [Lample and Conneau, 2019], fine-tuning which enables generalizing natural language understanding [Pires et al., 2019] and machine translation [Johnson et al., 2017] to languages without labeled data. Pre-training on multimodal speech is analogous to pre-training on bitext, while the former requires less manual effort to obtain by collecting more measurements when speech is produced. In addition, time-synchronicity between modalities provides additional self-supervision, enabling u-HuBERT to more easily learn frame-level modality-agnostic features.

## 3 Method

### 3.1 Background: audio-visual HuBERT

We build our framework upon AV-HuBERT [Shi et al., 2022a], a self-supervised representation learning model designed to pre-train a single type of multimodal speech data: audio-visual speech. AV-HuBERT first encodes temporally aligned audio speech $X^a \in \mathbb{R}^{T_a \times C_a}$ and visual speech $X^v \in \mathbb{R}^{T_v \times C_v \times H \times W}$ into feature sequences $h^a \in \mathbb{R}^{T \times D_h}$ and $h^v \in \mathbb{R}^{T \times D_h}$ of the same lengths, which are then fused frame-by-frame $\texttt{FUSE}(h^a, h^v) \in \mathbb{R}^{T \times D_c}$ and processed by a Transformer backend [Vaswani et al., 2017] to produce contextualized audio-visual representations $c^{av} \in \mathbb{R}^{T \times D_c}$.

This model is trained by iterating two steps. First, a K-means clustering model is trained on some audio-visual features to produce frame-level targets $Y^{av} \in [1, \cdots, K]^T$, where $K$ is the number of clusters. Second, the AV-HuBERT model is trained with a masked cluster prediction task, where spans from the audio-visual input are randomly masked and the model learns to predict the clusters of the masked frames given the context. The model is forced to learn the underlying linguistic structure of audio-visual speech to infer the content in the unseen regions based on the context. The intermediate feature of a trained model, which is better than previous iteration, is used to produce targets for the subsequent iteration.

### 3.2 Self-supervised pre-training with many modalities

We extend the framework to utilize not only multimodal data, but also unimodal data that can be more easily obtained in large quantities. The main idea is to apply modality dropout to multimodal

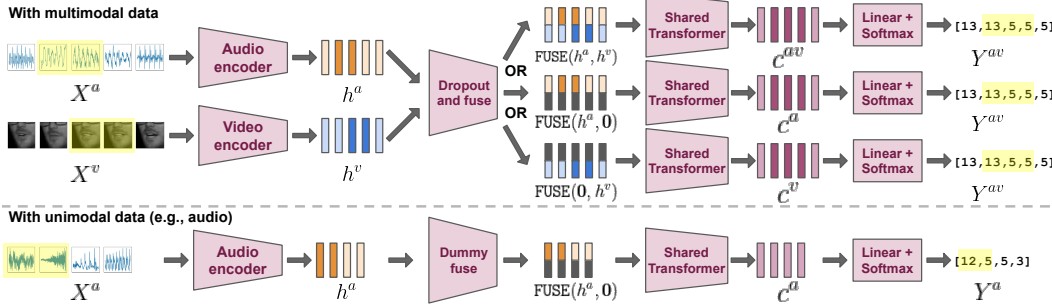

**Pre-training (predict pseudo labels of highlighted frames)**

Figure 2: u-HuBERT pre-trains on multimodal and unimodal speech with a unified objective: predicting cluster assignments of the frames that are masked at the input (highlighted by yellow shades). For multimodal data, a subset of modalities are randomly selected to predict the same targets.

speech along with a shared Transformer [Vaswani et al., 2017] to learn modality-agnostic features that can be quantized by a shared codebook to produce pseudo labels for masked prediction.

Given unimodal and multimodal speech data from many modalities, we assume that there is an anchor modality that is paired with all other modalities. For simplicity, assume the anchor modality is audio, and we have unlabeled audio, visual, and audio-visual speech. In the first iteration, because a modality-agnostic feature extractor is not available yet, we cluster the anchor modality features to produce targets for unimodal and multimodal speech that contain the anchor modality. Concretely speaking, this produces pseudo paired data for audio speech $X^a$ and audio-visual speech $(X^a, X^v)$ with audio-based cluster assignment $Y^a$ as the target. Like AV-HuBERT, each modality has its own feature extractor: $f^a$ for audio and $f^v$ for video, and feature sequences are fused frame-by-frame before being passed onto the Transformer $g$ to produce contextualized features $c^{av} = g(\text{FUSE}(f^a(X^a), f^v(X^v)))$. When a modality is absent, its feature sequence is simply replaced by zero vectors, for example, $c^a = g(\text{FUSE}(f^a(X^a), \mathbf{0}))$. Following HuBERT [Hsu et al., 2021a], this model is trained with a masked prediction objective.

To encourage the model to learn modality-agnostic features, modality dropout is applied to multimodal data to randomly drop a subset of the modalities, effectively creating multiple copies of the data with the same target but different input modalities, as depicted in the top half of Figure 2. While modality dropout was also used in AV-HuBERT, it was implemented for a different reason – to reduce the discrepancy between pre-training and fine-tuning for unimodal downstream tasks, which resulted in only minor improvement (reducing the WER from 57.0% to 55.3% as shown in Table D.1 in Shi et al. [2022a]). In contrast, we will demonstrate that modality dropout is essential for mixed modality pre-training and zero-shot generalization.

After the first iteration, the model can be used to extract modality-agnostic features for all unimodal and multimodal data, which are clustered to produce the pseudo label for next iteration of pre-training as illustrated in Figure 1. For multimodal data like audio-visual speech, one can produce pseudo labels using features extracted from the complete multimodal input (e.g., $Y^{av}$ from audio-visual input), or those extracted from input with a subset of modalities (e.g., $Y^v$ using only the video stream). We opt for the former and study the impact of these choices in Secton 4.2.

### 3.3 Multimodal and unimodal fine-tuning with partial modalities

Once the model is pre-trained, we remove the cluster prediction head (the linear layer on top of transformer used to predict cluster assignment distribution) and add a downstream task-specific prediction head. The pre-trained model can be fine-tuned on labeled multimodal speech, unimodal speech, or speech with mixed modalities. More importantly, the modalities included in the fine-tuning data do not necessarily cover the modalities seen during pre-training, yet the fine-tuned model can still handle the downstream task on all pre-trained modalities, which we refer to as *zero-shot modality generalization*. This is because the pre-trained model has successfully learned modality-agnostic representations, where same speech manifested in different modalities are mapped to similar

representations. Thus mapping from the representation of one modality (e.g., audio) to a target output (e.g., text) can be directly applied to other pre-trained modalities (e.g., video).

Similar to mBART [Liu et al., 2020] for zero-shot machine translation, catastrophic forgetting [Lee et al., 2019, Üstün et al., 2021] may occur during fine-tuning and hence hinders its generalization across modalities. Particularly, this is because the distributions of Transformer input and output at earlier layers are less modality-invariant. To alleviate the issue, we apply modality dropout when multimodal data is used, and explore a few techniques to prevent the representations from drifting too much, including freezing the entire pre-trained model or freezing a number of initial layers.

## 4 Experiments

### 4.1 Pre-training setup

We evaluate u-HuBERT on audio, visual, and audio-visual speech, where larger quantities of unlabeled and labeled data are available. Two audio-visual and one audio-only datasets are used for pre-training: (1) **LRS3** [Afouras et al., 2018] with 433 hours of English audio-visual speech, (2) **VoxCeleb2-En (VC2-En)** with 1,326 hours of English YouTube audio-visual speech filtered from VoxCeleb2 [Chung et al., 2018] by Shi et al. [2022a], and (3) **TED-LIUM release 3 (TD)** [Hernandez et al., 2018] with 452 hours of English audio collected from the same domain as LRS3. To improve noise robustness, we apply online noise augmentation where each utterance is corrupted with additive noise at 0dB sampled from MUSAN [Snyder et al., 2015] with a probability of 0.25. More details can be found in Appendix A. These datasets are solely used to perform experiments for the purpose of comparing with previous work in a fair setup.

We adopt the AV-HuBERT-LARGE model Shi et al. [2022a][3]: the video encoder is a modified ResNet-18 model [He et al., 2016] that takes image frames of 88x88 pixels sampled at 25Hz at input and produce a sequence of 512 dimensional embeddings at the same frame rate; audio is represented as filter bank features sampled at 100 Hz, stacked every four frames to match the video frame rate, and projected with a linear layer to 1024-D features. FUSE concatenates features frame-by-frame and linearly projects them to the embedding dimension of the shared Transformer. The shared Transformer has 24 layers, with 16 heads, embedding dimension 1024, and FFN dimension 4096, using the pre-norm residual connection setup [Nguyen and Salazar, 2019].

To directly compare with the publicly available fifth iteration noise-augmented AV-HuBERT-LARGE model trained on LRS3 and VC2-En, the same feature extractor (fourth iteration BASE model) and the same codebook (learned from multimodal features $C^{av}$) are used to generate pseudo labels for multimodal LRS3 and VC2-En and unimodal TD. We then pre-train u-HuBERT on the combined data for 1M updates on 64 32GB V100 GPUs using the Adam optimizer [Kingma and Ba, 2015] and a learning rate of 0.002. Gradient norm is clipped at 1.0. to stablize training. A batch size of maximal 40 seconds per GPU is used. When pre-trained on multimodal data, audio and video are dropped with a probability of 0.25 and 0.25, respectively.

### 4.2 Learning modality-agnostic features and codebooks with modality dropout

We argue that modality dropout leads to learning modality-agnostic representations through forcing the model to predict the same target with different subsets of input modalities given multimodal input, which in turn enables construction of a shared codebook for all modalities. To verify this, we compare two models that are identical except for whether modality dropout is activated: the public noise-augmented AV-HuBERT LARGE model trained on LRS3 and VC2-En with modality dropout and its counterpart. For each model, we infer audio, visual, and audio-visual features for each LRS3 validation utterances, randomly sample 500 frames, pool the features, and run t-SNE [Van der Maaten and Hinton, 2008] dimension reduction for visualization (Figure 3). It is visually evident that without modality dropout, features extracted from different modalities have different distributions (bottom row), contrary to those from the model with modality dropout (top row).

We next adopt the phone normalized mutual information (PNMI) metric presented in Hsu et al. [2021a] to quantify cluster quality. For each model, we create four codebooks by clustering audio-

---

[3]`https://github.com/facebookresearch/av_hubert/blob/main/avhubert/conf/pretrain/noise_large_vox_iter5.yaml`

visual features $C^{av}$, audio features $C^a$, visual features $C^v$, or all three combined $\cup_m C^m$, quantize each type of features with each codebook, and measure the normalized mutual information $\in [0, 1]$ between frame-level phone labels and cluster assignment (Table 1). Because features inferred from different modalities exhibit similar distributions with modality dropout, all four codebooks lead to similar PNMI for a given feature. In contrast, without modality dropout, cross-quantization (e.g., $Y^{av}$ with $C^v$ codebook) often leads to significantly worse PNMI, and PNMI is worse for all *(codebook, feature)* combinations compared with that from the modality dropout model. Note for a given codebook, the PNMI still differs among modalities which is due to the discrepancy of information carried per modality.

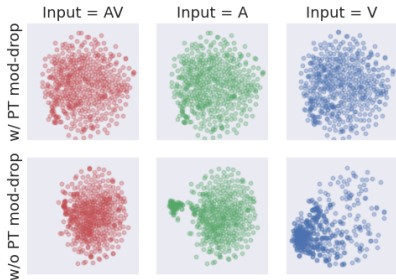

Figure 3: t-SNE projection of AV/A/V speech representations learned by models differing in modality dropout.

| K-means training data | PNMI ($\uparrow$) | | | | | |
| | w/ PT mod-drop | | | w/o PT mod-drop | | |
| | $Y^{av}$ | $Y^a$ | $Y^v$ | $Y^{av}$ | $Y^a$ | $Y^v$ |
|---|---|---|---|---|---|---|
| $\cup_m C^m$ | 43.60 | 43.72 | 38.01 | 41.89 | 36.59 | 21.78 |
| $C^{av}$ | 42.89 | 42.96 | 36.89 | 42.83 | 38.46 | 26.17 |
| $C^a$ | 43.18 | 43.28 | 37.11 | 42.87 | 37.28 | 20.99 |
| $C^v$ | 42.23 | 42.57 | 37.68 | 6.85 | 13.68 | 23.54 |

Table 1: Cluster quality (PNMI) of each modality $Y^{av}/Y^a/Y^v$ using codebooks learned from features of different modalities, obtained from models pre-trained with and without modality dropout.

## 4.3 Speech recognition

**Fine-tuning setup**  LRS3 *trainval* and *pretrain*, combined for 433 hours, are used for training, with the same 1,200 utterances as Shi et al. [2022a] split for validation. Results on the *test* split are reported. Noise augmentation with MUSAN used in pre-training is also adopted during fine-tuning. A 9-layer randomly initialized Transformer decoder with 8 attention heads and 1024/4096-D embedding/FFN is appended to the pre-trained u-HuBERT, and the entire model is fine-tuned with a cross-entropy loss predicting the next text token given the input and the previous text tokens. Text is represented as unigram-based subword units [Kudo, 2018] with a vocabulary size of 1,000. Modality dropout is applied when fine-tuning on multimodal input. Additionally, we sweep learning rate, number of updates, steps to freeze, and layers to freeze throughout fine-tuning. We evaluate each model, regardless of the modality used for fine-tuning, on all kinds of input modalities and report the word error rates (WERs) without fusing external language models. To gauge the robustness to noise, we follow the same protocol in Shi et al. [2022b] to noisy audio and audio-visual test sets by adding babble noise at 0 decibel (dB) to the audio stream for evaluation. Detailed hyperpamater setups can be found in Appendix A, and more ablation studies can be found in Appendix B.1.

**Modality dropout is essential for building a single model**  Table 2 compares fine-tuning from scratch, as well as from pre-trained models with and without modality dropout, using labeled audio, visual, or audio-visual speech, and highlights the results of zero-shot modality transfer where labeled data in test modality are not used. First, models without pre-training are significantly worse. Second, when fine-tuning on audio-visual data without pre-training modality dropout, performance drops significantly on unimodal data if pre-training modality dropout is not applied (e.g., 1.7% to 21.4% on clean audio). This can be alleviated with fine-tuning modality dropout, improving the average WER from 24.5% to 12.3%, but is still behind applying it in both pre-training and fine-tuning (11.4% average). We also note that fine-tuning dropout leads to consistent improvement on noisy AV test set, likely because occasionally dropping the audio stream encourages the model to rely more on the visual input which improves noise robustness.

The benefit of applying pre-training modality dropout is magnified in unimodal fine-tuning: the model fine-tuned only on labeled audio data yields the same performance on clean and noisy audio-visual input (1.3% and 4.6%) as the model fine-tuned with labeled audio-visual data, while being slight behind on video input compared to the model fine-tuned with labeled video data (31.6% vs. 28.7%). When it is compared to the model pre-trained without modality dropout and fine-tuned also with labeled audio, huge gains are observed on zero-shot scenarios, for example 96.8% versus 31.6%

Table 2: Speech recognition results on LRS3 test. PT indicates if a model is pre-trained (on LRS3 and VC2-En). PT mod-drop-p and FT-drop-p denote if modality dropout is applied in pre-training and fine-tuning, respectively. FT mod denotes the labeled data used for fine-tuning: audio-visual (AV), audio (A), or visual (V). Zero-shot scenarios are highlighted with blue shades.

| PT | PT mod-drop | FT mod | FT mod-drop | AV-WER Clean | AV-WER Noisy | A-WER Clean | A-WER Noisy | V-WER | Avg-WER |
|---|---|---|---|---|---|---|---|---|---|
| *fine-tuned on 433h* | | | | | | | | | |
| ✗ | n/a | AV | ✗ | 3.8 | 17.2 | 28.2 | 87.6 | 83.4 | 44.0 |
| ✓ | ✗ | AV | ✗ | 1.3 | 4.8 | 21.4 | 52.6 | 42.3 | 24.5 |
| ✓ | ✓ | AV | ✗ | 1.2 | 5.2 | 1.7 | 25.5 | 32.4 | 13.2 |
| ✗ | n/a | AV | ✓ | 3.6 | 15.9 | 4.6 | 44.8 | 63.7 | 26.5 |
| ✓ | ✗ | AV | ✓ | 1.3 | 4.1 | 1.8 | 23.1 | 31.0 | 12.3 |
| ✓ | ✓ | AV | ✓ | 1.3 | 4.6 | 1.5 | 20.5 | 29.1 | 11.4 |
| ✗ | n/a | A | n/a | ✗ | ✗ | 4.0 | 37.3 | ✗ | ✗ |
| ✓ | ✗ | A | n/a | 1.5 | 18.0 | 1.6 | 20.9 | 96.8 | 27.8 |
| ✓ | ✓ | A | n/a | 1.3 | 4.6 | 1.4 | 19.3 | 31.6 | 11.6 |
| ✗ | n/a | V | n/a | ✗ | ✗ | ✗ | ✗ | 60.3 | ✗ |
| ✓ | ✗ | V | n/a | 11.3 | 21.8 | 80.3 | 97.7 | 28.0 | 47.8 |
| ✓ | ✓ | V | n/a | 2.1 | 5.1 | 2.3 | 20.9 | 28.7 | 11.8 |

V-WER and 18.0% versus 4.6% noisy AV-WER. Similar trend can be observed when fine-tuning with labeled visual speech.

**Zero-shot modality transfer is not achieved through memorization** Because the dataset used for fine-tuning is a subset of the pre-training data, there could be suspicions that the model memorizes the audio-visual pairs from pre-training and associates them with the labeled unimodal data to enable zero-shot transfer. To avoid cheating, we fine-tune our model on LibriSpeech [Panayotov et al., 2015], an audio-only read audiobook dataset that is out-of-domain relative to the pre-training data composed of oratory talks (Table 3). Despite the degradation compared to fine-tuning on in-domain LRS3 data, the model still performs decently on zero-shot transfer scenarios compared to models pre-trained without modality dropout in Table 2 and 3.

Table 3: Fine-tuning on out-of-domain labeled audio speech that does not overlap with the pre-training data, and evaluating on LRS3 *test*. Zero-shot scenarios are highlighted with blue shades.

| PT data | PT mod-drop | FT data | FT mod | AV (Clean) | AV (Noisy) | A (Clean) | A (Noisy) | V |
|---|---|---|---|---|---|---|---|---|
| LRS3+VC2-En | ✗ | LibriSpeech | A | 6.2 | 11.0 | 20.9 | 65.9 | 64.7 |
| LRS3+VC2-En | ✓ | LibriSpeech | A | 4.5 | 8.9 | 4.8 | 27.6 | 39.4 |

**Comparison with state-of-the-art models** Table 4 compares u-HuBERT with models from the literature, all of which are modality-specific. In contrast, u-HuBERT is the only single model that can be fine-tuned on *any* modality and test on *all* modalities while achieving performance on par or better than the best modality-specific models using similar amounts of labeled data. Furthermore, compared to AV-HuBERT, u-HuBERT can be pre-trained on additional unimodal data to yield even better performance.

## 4.4 Speech translation

We repeat the experiment on speech translation to study if u-HuBERT enables construction of a single model and achieves zero-shot generalization on a different task.

**Fine-tuning setup** To simulate speech translation setting, we translate the English transcriptions into Spanish to create paired (English audio, English video, Spanish text) data. The 30 hour LRS3 trainval split is used for fine-tuning. We use the same decoder for speech translation as for speech recognition, where a 9-layer randomly initialized Transformer is appended to the pretrained u-HuBERT encoder to decode English speech into Spanish text. Unigram-based sentencepiece units with a vocabulary

Table 4: Comparison with state-of-the-art audio, visual, and audio-visual speech recognition results reported on LRS3. Zero-shot scenarios are highlighted with blue shades. u-HuBERT pre-trained without unlabeled unimodal data is effectively AV-HuBERT but fine-tuned with hyperparameters optimized for performance on all three modalities. [†] refers to *filtered* public datasets where the list of used utterances are not public. [‡] refers to non-public datasets.

| Method | Unlab data | | Mod | Lab data | WER (%) | | |
| | AV (hr) | A (hr) | | Dataset (hr) | AV | A | V |
|---|---|---|---|---|---|---|---|
| ***supervised learning*** | | | | | | | |
| Ma et al. [2021] | - | - | AV | LRW, LRS3 (590) | 2.3 | ✗ | ✗ |
| Ma et al. [2021] | - | - | A | LRW, LRS3 (590) | ✗ | 2.3 | ✗ |
| Ma et al. [2021] | - | - | V | LRW, LRS3 (590) | ✗ | ✗ | 43.3 |
| Xu et al. [2020] | - | - | AV | LRW, LRS3 (590) | 6.8 | ✗ | ✗ |
| Ma et al. [2022] | - | - | V | LRW, LRS2, LRS3, AVSpeech[†] (1,459) | ✗ | ✗ | 31.5 |
| Afouras et al. [2021] | - | - | V | LRS2, LRS3, MV-LRS[‡], TEDx$_{ext}^{\ddagger}$ (2,676) | ✗ | ✗ | 30.7 |
| Makino et al. [2019] | - | - | AV | YT31k[‡] (31,000) | 4.5 | ✗ | ✗ |
| Makino et al. [2019] | - | - | A | YT31k[‡] (31,000) | ✗ | 4.8 | ✗ |
| Makino et al. [2019] | - | - | V | YT31k[‡] (31,000) | ✗ | ✗ | 33.6 |
| Serdyuk et al. [2021] | - | - | V | YT90k[‡] (90,000) | ✗ | ✗ | 25.9 |
| Serdyuk et al. [2021] | - | - | V | LRS3, YT90k[‡] (90,433) | 2.3 | ✗ | ✗ |
| ***semi-supervised and self-supervised learning*** | | | | | | | |
| [Shi et al., 2022b] | LRS3, VC2-En (1,759) | - | AV | LRS3 (433) | 1.4 | ✗ | ✗ |
| [Shi et al., 2022b] | LRS3, VC2-En (1,759) | - | A | LRS3 (433) | ✗ | 1.6 | ✗ |
| [Shi et al., 2022a] | LRS3, VC2-En (1,759) | - | V | LRS3 (433) | ✗ | ✗ | 28.6 |
| Afouras et al. [2020] | VC2-clean[†] (334) | - | V | LRW, LRS3 (590) | ✗ | ✗ | 59.8 |
| u-HuBERT | LRS3, VC2-En (1,759) | - | AV | LRS3 (433) | 1.3 | 1.5 | 29.1 |
| u-HuBERT | LRS3, VC2-En (1,759) | - | A | LRS3 (433) | 1.3 | 1.4 | 31.6 |
| u-HuBERT | LRS3, VC2-En (1,759) | - | V | LRS3 (433) | 2.1 | 5.1 | 28.7 |
| u-HuBERT | LRS3, VC2-En (1,759) | TD (452) | AV | LRS3 (433) | 1.2 | 1.4 | 27.2 |

size of 1,000 are used as decoding output units. The overall fine-tuning and decoding paradigms remain same as speech recognition. Detailed hyperparameters can be found in Appendix A. BLEU score Papineni et al. [2002] is used to evaluate the translation model.

**Translation performance** The performance of u-HuBERT on speech translation is shown in Table 5 [4]. As is similar to the observation in speech recognition, the model hardly produces meaningful results if its encoder is randomly initialized (Avg-BLEU: 3.3). u-HuBERT pre-training not only greatly improves translation quality by roughly 10 fold (Avg-BLEU: 3.3→35.0), but also shows strong performance in zero-shot modality transfer. For instance, the model fine-tuned with audio-only speech achieves 34.3 BLEU score when it is tested using audio-visual speech under noisy environment, outperforming its audio-only counterpart (26.5). In tasks such as speech translation, where labeled multimodal data can be difficult to harvest, our pre-training approach shows potential in enhancing the noise robustness supplied by the extra visual modality at test time even without exposure to any labeled visual data during training. Within u-HuBERT, modality dropout is an essential component for acquiring such capacity, as is suggested by improvement in average BLEU score it brings (A: 27.3→34.7, V: 17.0→31.7). In cases with available labeled multimodal data, a single audio-visual model is able to achieve performance superior or on par to a modality-specific model (A: 42.2 vs. 42.9, V: 27.6 vs. 26.4).

## 4.5 Ablation studies

**Pre-training modality dropout parameters** To better understand the impact of pre-training modality dropout configurations on zero-shot modality transfer, we conducted pre-training ablation studies with a reduced setup: for each configuration, a BASE model is pre-trained on 433 hours of LRS3 data using 32 GPUs for 200K updates and then fine-tuned on 30 hours of LRS3 labeled data (the "trainval" split). Four modality dropout configurations are considered, represented by the probability of using both streams ($p_{av}$), audio stream ($p_a$) and video stream ($p_v$) in Table 6. The first configuration ($p_{av}, p_a, p_v$) = (1.00, 0.00, 0.00) is equivalent to not using modality dropout.

When fine-tuning on audio-visual or audio-only data, among the models pre-trained with modality dropout, setting the $p_v$ higher leads to slightly worse performance on audio-visual and audio-only test sets. In contrast, when fine-tuning on visual-only data, setting the probability of V-only higher leads

---

[4]The results of models finetuned with 433 hours of labeled speech can be found in Table B9 in the Appendix.

Table 5: Speech translation results on LRS3 test. Headers are the same as Table 2.

| PT | PT mod-drop | FT mod | FT mod-drop | AV-BLEU Clean | AV-BLEU Noisy | A-BLEU Clean | A-BLEU Noisy | V-BLEU | Avg-BLEU |
|---|---|---|---|---|---|---|---|---|---|
| *fine-tuned on 30h* | | | | | | | | | |
| ✗ | n/a | AV | ✓ | 3.6 | 5.6 | 3.5 | 2.4 | 1.3 | 3.3 |
| ✓ | ✗ | AV | ✓ | 39.5 | 34.8 | 39.2 | 24.5 | 25.1 | 32.6 |
| ✓ | ✓ | AV | ✓ | 42.4 | 36.5 | 42.2 | 26.0 | 27.6 | 35.0 |
| ✗ | n/a | A | n/a | ✗ | ✗ | 7.2 | 4.0 | ✗ | ✗ |
| ✓ | ✗ | A | n/a | 38.3 | 26.8 | 38.0 | 22.9 | 10.8 | 27.3 |
| ✓ | ✓ | A | n/a | 43.4 | 34.3 | 42.9 | 26.5 | 26.3 | 34.7 |
| ✗ | n/a | V | n/a | ✗ | ✗ | ✗ | ✗ | 0.5 | ✗ |
| ✓ | ✗ | V | n/a | 24.9 | 20.8 | 16.6 | 7.2 | 15.2 | 17.0 |
| ✓ | ✓ | V | n/a | 38.1 | 33.2 | 37.1 | 23.1 | 26.4 | 31.7 |

to slightly better average performance. Nevertheless, all three configurations achieve significantly better performance compared to the model pre-trained without modality dropout, confirming that activating modality dropout is the key to successful zero-shot modality transfer.

Table 6: Ablation study for pre-training modality dropout probabilities with a reduced training setup.

| PT $p_{av}$ | PT $p_a$ | PT $p_a$ | FT mod | FT mod-drop | AV-WER Clean | AV-WER Noisy | A-WER Clean | A-WER Noisy | V-WER | Avg-WER |
|---|---|---|---|---|---|---|---|---|---|---|
| 1.00 | 0.00 | 0.00 | AV | ✗ | 5.54 | 14.57 | 17.83 | 67.66 | 59.72 | 33.06 |
| 0.70 | 0.15 | 0.15 | AV | ✗ | 5.24 | 15.11 | 6.21 | 47.42 | 50.65 | 24.93 |
| 0.50 | 0.25 | 0.25 | AV | ✗ | 5.17 | 15.36 | 6.45 | 48.59 | 51.42 | 25.40 |
| 0.50 | 0.15 | 0.35 | AV | ✗ | 6.04 | 17.59 | 7.46 | 51.78 | 51.06 | 26.79 |
| 1.00 | 0.00 | 0.00 | A | n/a | 6.26 | 15.47 | 10.67 | 60.81 | 66.89 | 32.02 |
| 0.70 | 0.15 | 0.15 | A | n/a | 5.83 | 14.23 | 6.10 | 41.29 | 52.13 | 23.92 |
| 0.50 | 0.25 | 0.25 | A | n/a | 5.50 | 14.84 | 5.96 | 41.72 | 48.72 | 23.35 |
| 0.50 | 0.15 | 0.35 | A | n/a | 6.22 | 16.00 | 6.54 | 43.71 | 48.71 | 24.24 |
| 1.00 | 0.00 | 0.00 | V | n/a | 17.45 | 26.68 | 27.21 | 79.01 | 66.65 | 43.40 |
| 0.70 | 0.15 | 0.15 | V | n/a | 13.28 | 20.23 | 13.59 | 48.30 | 52.40 | 29.56 |
| 0.50 | 0.25 | 0.25 | V | n/a | 11.01 | 18.18 | 11.54 | 47.96 | 49.19 | 27.58 |
| 0.50 | 0.15 | 0.35 | V | n/a | 10.69 | 18.74 | 11.69 | 50.10 | 49.11 | 28.07 |

**Pre-training dynamics with additional unimodal data**    While mixed modality pre-training allows the model to be trained on larger quantities of data, it also dilutes the in-domain data with respect to audio-visual or other unimodal tasks, which may lead to worse performance on those tasks. We pre-train on audio-visual data and mixed-modal data (audio-visual and audio-only) respectively for different numbers of updates and study if degradation occurs to audio-visual and visual speech recognition in the mixed-modal case.

The baseline (PT on AV) is pre-trained on audio-visual LRS3, and the proposed mixed-modal pre-training (PT on AV+A) uses the same audio-visual data as the baseline plus the VC2-En audio data. For each setup we pre-train the model for {100, 200, 400, 600}K updates. Results in Figure 4 showed that adding unlabeled audio data consistently improves the audio speech recognition performance at all number of updates. In contrast, the performance of "PT on AV+A" lags behind "PT on A" on audio-visual and visual speech recognition when a model is trained for fewer than 400K updates. Nevertheless, when a model is trained for longer (600K updates), mixed-modal pre-training outperforms the baseline on all three tasks. The observation suggests that u-HuBERT can benefit from training on more data and continue improving with more updates, while the performance of the baseline saturates earlier due to the lack of data.

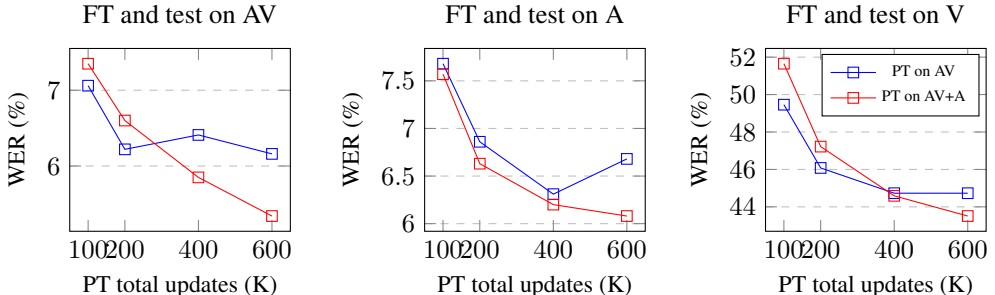

Figure 4: Ablation study comparing pre-training on audio-visual data (LRS3) versus on both audio-visual data (LRS3) and audio-only data (audio from VC2-En) for different numbers of steps.

## 5 Discussion

We assume that there is an anchor modality where all other modalities appear in some multimodal data paired with it. This enables the model to predict shared targets and learn modality-agnostic representations for all modalities, such that non-anchor unimodal data can be used in the subsequent iterations. While this may seem limiting, most multimodal speech are paired with audio [Richard et al., 2021, Hueber et al., 2010, Richmond et al., 2011, Livescu et al., 2009] since audio is the primary measurement for speech and other modalities are often supplementary. This makes the assumption of having audio as the anchor modality practical. Furthermore, the idea can be easily extended to settings where any modality can be connected to the anchor via some intermediate modality. In short, a modality that is $N$-hop away from the primary anchor can be used in the $(N + 1)$-th iteration.

The only case not supported u-HuBERT is pre-training on modalities that cannot be connected through any multimodal data, for example, on a collection of audio, visual, audio-visual and EMG speech. A shared codebook cannot be derived for EMG and the rest of the modalities, and it is unclear if the model can still benefit from joint training. The setup has also been found much more challenging on other pairs of modalities: for example, SLAM Bapna et al. [2021] shows that without paired text and speech data, joint pre-training on unlabeled speech and text degrades the performance compared to a model pre-trained only on speech. We leave such exploration to future work. In addition, we also include ethical discussion in Appendix C.

## 6 Conclusion

Recent studies in applied machine learning trend toward building things that are *general*, for example, general perception module [Jaegle et al., 2021], general self-supervised objective [Baevski et al., 2022], single model for many tasks [Brown et al., 2020, Kaplan et al., 2020, Aghajanyan et al., 2022] and/or many input [Girdhar et al., 2022, Hu and Singh, 2021], instead of optimizing for performance on very specific setups. Our work contributes to this line of research by presenting a unified self-supervised objective to pre-train a model on speech of many modalities. Specifically, it can be fine-tuned for many different speech tasks on labeled data in any combination of pre-trained modalities, resulting in a single model that can process all combinations of pre-trained modalities to perform the fine-tuned task.

We envision our work can bring substantial benefit to multimodal speech processing. Given there are significantly more unlabeled audio compared to unlabeled multimodal speech, being able to utilize unlabeled audio can greatly improve the quality of multimodal speech representations. Furthermore, there are few or even no labeled multimodal speech for many tasks, such as speech translation. Hence, it is essential to build a model capable of zero-shot input modality transfer to perform these tasks with modalities without labeled data.

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
