# A Experiment Details

## A.1 Data

Table A1 summarizes the datasets used in this paper, which are all licensed under CC BY-NC-ND or CC BY and have been used extensively by the research communities. Speech datasets are sourced from interviews, TED talks, and audiobooks, which are not expected to contain offensive content.

Table A1: Dataset specifications

| Dataset | Type | Size (hr) | Source | License |
|---|---|---|---|---|
| LRS3 v0.4 [Afouras et al., 2018] | audio-visual speech | 433 | TED and TEDx | CC BY-NC-ND 4.0 |
| VoxCeleb1 [Nagrani et al., 2017] | audio-visual speech | 352 | Interviews on YouTube | CC BY-NC-ND 4.0 |
| VoxCeleb2 [Chung et al., 2018] | audio-visual speech | 2,794 | Interviews on YouTube | CC BY-NC-ND 4.0 |
| TED-LIUM 3 [Hernandez et al., 2018] | audio speech | 452 | TED | CC BY-NC-ND 3.0 |
| LibriSpeech [Panayotov et al., 2015] | audio speech | 960 | LibriVox audiobooks | CC BY 4.0 |
| MUSAN [Snyder et al., 2015] | music / speech / noise | 109 | US Public Domain / under CC | CC BY 4.0 |

## A.2 Fine-tuning

Table A2 summarizes the hyperparameters used for ASR fine-tuning.

Table A2: Speech recognition fine-tuning hyperparameters

| FT mod | AV | A | V |
|---|---|---|---|
| batch size | 40 sec | 40 sec | 40 sec |
| # GPU | 8 | 8 | 8 |
| audio dropout | 0.5 | n/a | n/a |
| video dropout | 0.5 | n/a | n/a |
| learning rate | 4e-4 | 4e-4 | 5e-4 |
| LR phase ratio | [0.33, 0, 0.67] | [0.33, 0, 0.67] | [0.33, 0, 0.67] |
| update steps | 60,000 | 60,000 | 60,000 |
| freezing step | 30,000 | 30,000 | 30,000 |
| freezing layers | 18 | 8 | 6 |

Table A3 summarizes the hyperparameters used for speech translation.

Table A3: Speech translation fine-tuning hyperparameters

| FT mod | AV | A | V |
|---|---|---|---|
| batch size | 40 sec | 40 sec | 40 sec |
| # GPU | 8 | 8 | 8 |
| audio dropout | 0.5 | n/a | n/a |
| video dropout | 0.5 | n/a | n/a |
| learning rate | 0.001 | 0.001 | 0.001 |
| LR phase ratio | [0.33, 0, 0.67] | [0.33, 0, 0.67] | [0.33, 0, 0.67] |
| update steps | 30,000 | 30,000 | 30,000 |
| freezing step | 24,000 | 24,000 | 24,000 |
| freezing layers | all | all | all |

## A.3 Speech recognition decoding

Beam search decoding is used with a length weight $\alpha$, which searches for the hypothesis $z_{1:T}$ that maximizes

$$\frac{\sum_{t=1}^{T} P(z_t|z_{1:t-1}, X)}{T^\alpha} \tag{1}$$

For the results in the main paper, we do grid search from beam size $\in \{1, 5, 10, 15, 20, 25\}$ and $\alpha \in \{0, 0.5, 1.0, 1.5\}$. For the ablation studies in the appendix a beam size of 10 and $\alpha = 1.0$ is used.

# B  Extend Experimental Results

## B.1  Impact of fine-tuning hyperparameters

We conduct the ablation studies in this section with the models pre-trained on multimodal LRS3 and VC2-En. By default, the one pre-trained with modality dropout is used.

### B.1.1  Fine-tuning on multimodal data

Table B4 shows how fine-tuning modality dropout configurations affect ASR performance. Models fine-tuned with dropout can yield better performance on all input modality than the one without (reported in the caption). When setting m-drop-p and a-drop-p within the range of $[0.25, 0.75]$, the results have limited variation ($[1.43\%, 1.65\%]$ WER for AVSR, $[1.82\%, 2.22\%]$ WER for ASR, and $[28.85\%, 30.88\%]$ WER for VSR). Similarly setting the values to 0.5 yields reasonable performance for all modalities.

Table B4: Impact of fine-tuning modality dropout for audio-visual speech recognition. Without modality dropout, AV-WER/A-WER/V-WER = 1.48/2.43/34.37.

| FT mod | m-drop-p | AV-WER | | | A-WER | | | V-WER | | |
|---|---|---|---|---|---|---|---|---|---|---|
| | a-drop-p= | 0.25 | 0.50 | 0.75 | 0.25 | 0.50 | 0.75 | 0.25 | 0.50 | 0.75 |
| AV | 0.25 | 1.58 | 1.46 | 1.61 | 2.04 | 2.01 | **1.82** | 30.88 | 30.80 | 30.05 |
| AV | 0.50 | **1.43** | 1.57 | 1.53 | 1.92 | 2.00 | 1.85 | 30.34 | 29.73 | **28.85** |
| AV | 0.75 | 1.53 | 1.71 | 1.65 | 2.22 | 2.08 | 2.12 | 29.72 | 29.43 | 29.88 |

### B.1.2  Fine-tuning on unimodal data

Next, we study the impact of hyperparameters when fine-tuning on unimodal data. Different from fine-tuning on multimodal data, fine-tuning on unimodal data is more prone to catastrophic forgetting, leading to huge performance degradation on modalities unseen during fine-tuning. Hence, we focus on studying the impact from three hyperparameters:

- $L_{frz}$: number of u-HuBERT Transformer layers that are frozen throughout fine-tuning.
- $N_{frz}$: number of the fine-tuning updates where the entire u-HuBERT is frozen. After this many updates, the layers above ($N_{frz}$)-th layer are optimized jointly with the prediction head.
- *LR*: learning rate.

**Number of frozen layers** ($L_{frz}$)  Setting this effectively treats the first $L_{frz}$ layers as a fixed feature extractor, while the layers above are considered pre-trained and jointly optimized with the added prediction head during fine-tuning. Results are shown in Table B5.

When fine-tuned on audio speech from `PT mod-drop`, not freezing any layer leads to worse performance for all modalities, but freezing all the layers also reduces the model capacity and harms the performance for the fine-tuned modality and for audio-visual speech. For visual speech, the audio fine-tuned model achieves the best result when all the layers are frozen (30.60% compared to 35.68% when not freezing any layer), but the gap can be reduced from 5.08% to 1.23% when freezing 12 layers, in which audio-visual speech yields the best performance while audio is close to optimal.

When fine-tuned on visual speech from `PT mod-drop`, the model yields similar performance on visual speech regardless how many layers are frozen (with a spread of 0.5% WER). In contrast, results are optimal for both audio-visual and audio input when 12 layers are frozen, with a good trade-off between mod el capacity and representation distribution shift.

The trends are different when fine-tuning `PT no-mod-drop`. Because the model has only seen audio-visual input during pre-training, visual-only or audio-only input is considered out-of-distribution. Hence, when fine-tuning on visual speech with all the layers frozen ($L_{frz} = 24$), the performance on the fine-tuned input is significantly worse compared to fine-tuning `PT mod-drop` (45.18% versus 29.42%). Although this can be alleviated by unfreezing more layers, the model barely works for

audio-only input with WERs ranging from 21.55% to 54.27%. This again verifies that pre-training modality dropout is essential for achieving zero-shot modality transfer.

Table B5: Number of layers to freeze

| $L_{frz}$ | PT w/ mod-drop-p; FT on A | | | PT w/ mod-drop-p; FT on V | | | PT w/o mod-drop-p; FT on V | | |
| | AV-WER | A-WER | V-WER | AV-WER | A-WER | V-WER | AV-WER | A-WER | V-WER |
|---|---|---|---|---|---|---|---|---|---|
| 0 | 1.73 | 1.84 | 35.68 | 4.06 | 3.71 | 29.51 | 4.25 | 54.27 | **28.43** |
| 6 | 1.59 | **1.69** | 32.81 | 2.52 | 2.88 | 29.69 | 2.92 | 32.80 | 28.96 |
| 12 | **1.52** | 1.70 | 31.83 | **2.39** | **2.78** | 29.46 | **2.90** | 21.55 | 32.11 |
| 18 | 1.73 | 1.82 | 31.14 | 2.44 | 2.89 | 29.92 | 3.60 | 21.75 | 37.76 |
| 24 | 2.09 | 2.20 | **30.60** | 2.80 | 3.13 | **29.42** | 10.97 | 26.15 | 45.18 |

**Number of frozen steps** Results comparing the impact on freezing u-HuBERT for different numbers of fine-tuning steps are shown in Table B6. We can observe similar trends as those when varying the number of layers to freeze: not freezing at all leads worse performance on all input modalities, while treating the entire pre-trained model as a fixed feature extractor also leads to sub-optimal performance for the fine-tuned modality. Freezing for 30K updates out of 60K total updates leads to good balance for all modalities with `PT mod-drop`, which are 0.25%/0.11%/0.99% behind the optimal WERs when fine-tuned on audio, and 0%/0%/0.47% behind the optimal WERs when fine-tuned on visual speech. Similarly, the model pre-trained without modality dropout yields worse zero-shot modality transfer performance when fine-tuned on a single modality.

Table B6: Number of steps to freeze

| $N_{frz}$ | PT w/ mod-drop-p; FT on A | | | PT w/ mod-drop-p; FT on V | | | PT w/o mod-drop-p; FT on V | | |
| | AV-WER | A-WER | V-WER | AV-WER | A-WER | V-WER | AV-WER | A-WER | V-WER |
|---|---|---|---|---|---|---|---|---|---|
| 0 | 1.58 | 1.71 | 39.90 | 3.54 | 4.31 | 31.88 | 3.68 | 74.95 | 30.61 |
| 15K | **1.43** | **1.62** | 38.40 | 2.69 | 3.21 | 31.65 | 3.83 | 75.87 | 29.86 |
| 30K | 1.68 | 1.73 | 31.87 | **2.52** | **2.88** | 29.69 | **2.92** | 32.80 | **28.96** |
| 45K | 1.82 | 1.99 | 31.20 | 2.83 | 3.29 | **29.22** | 3.05 | **24.00** | 29.31 |
| 60K | 2.10 | 2.20 | **30.88** | 2.84 | 3.04 | 29.88 | 10.16 | 25.68 | 44.71 |

**Learning rate** Table B7 presents the results. When fine-tuning on audio speech, increasing learning rate leads to significantly worse WER on visual input (30.62% → 37.43%), but benefits audio-visual and audio speech up to $6 \times 10^{-4}$ and $8 \times 10^{-4}$, respectively, and the performance stays rather constant afterward. Similarly, when fine-tuning on visual speech, increasing learning rate to $10^{-3}$ hurts the result with audio input (2.73% → 3.41%); it also hurts the visual speech recognition performance, but the degradation is relatively minor (28.62% → 30.39%) compared to when fine-tuning on audio. In general, we observe a smaller learning is preferred for visual speech recognition, and increasing learning rates harms the zero-shot modalities more.

Table B7: Learning rate

| LR | PT w/ mod-drop-p; FT on A | | | PT w/ mod-drop-p; FT on V | | |
| | AV-WER | A-WER | V-WER | AV-WER | A-WER | V-WER |
|---|---|---|---|---|---|---|
| 2e-4 | 1.79 | 1.97 | **30.62** | 2.72 | 2.91 | **28.62** |
| 4e-4 | 1.68 | 1.73 | 31.87 | **2.45** | **2.73** | 29.25 |
| 6e-4 | **1.63** | 1.71 | 33.79 | 2.78 | 3.13 | 29.80 |
| 8e-4 | 1.65 | **1.67** | 35.29 | 2.63 | 3.23 | 29.97 |
| 10e-4 | 1.66 | 1.68 | 37.43 | 3.03 | 3.41 | 30.39 |

## B.2 Per-layer representation analysis

To better understand how u-HuBERT learns modality-agnostic features, we show the clustering quality of different layers per modality of a pre-trained u-HuBERT model. Similar to Table 1, we

report PNMI of layerwise clusters per modality quantized by audio-only ($C^a$), video-only ($C^v$), audio-visual ($C^{av}$) and all-combined ($\cup_m C^m$) codebook respectively (see Figure 5). When pre-trained without modality dropout, the model is unable to learn modality-agnostic features regardless of which layer to cluster, as can be shown from its overall lower PNMI of cross-modal clustering (e.g., PNMI of visual features quantized by $C^{av}$ codebook). To the model pre-trained with modality-dropout, the features of its intermediate layers become more agnostic to modality as the layer goes deeper, shown from the diminishing gap between its cross-modality and modality-specific performance in later layers. Furthermore, the approximately same quality of clusters from different codebooks towards the end suggests that the final layer is best suited to achieve a unified model that generalizes across modalities.

Figure 5: Comparison between models pre-trained with (left) and without (right) modality dropout in layerwise clustering quality (PNMI) of all modalities. A: audio, V: video, AV: audio-visual.

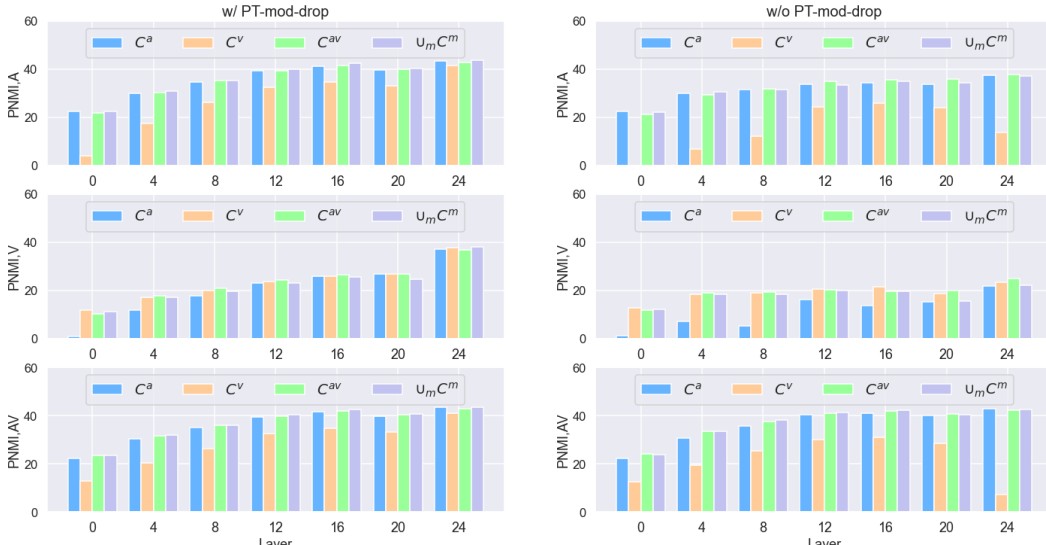

## B.3 Stability

Table B8 shows the WER mean and standard deviation over five runs when fine-tuning the two u-HuBERT models. We see that the one pre-trained additionally with unimodal audio data (TD) is significantly better on all input modalities.

Table B8: Variance of u-HuBERT performance fine-tuned on 433 hours of audio-visual speech.

| Unlab data | | AV-WER | | A-WER | | V-WER |
|---|---|---|---|---|---|---|
| AV | A | Clean | Noisy | Clean | Noisy | |
| LRS3+VC2-En | - | $1.34 \pm 0.08$ | $4.36 \pm 0.18$ | $1.47 \pm 0.04$ | $20.25 \pm 0.14$ | $29.57 \pm 0.29$ |
| LRS3+VC2-En | TD | $1.24 \pm 0.06$ | $3.33 \pm 0.10$ | $1.37 \pm 0.07$ | $15.43 \pm 0.09$ | $27.30 \pm 0.12$ |

## B.4 Speech Translation Results

Table B9 shows the performance of speech translation models fine-tuned with 433 hours of labeled data.

## C Ethical Discussion

The ability to build a model that can process unimodal or multimodal speech without needing labeled data in the target modality opens many possibilities for real world applications, since except for audio-only speech, labeled data are extremely lacking for other speech modalities. In particular, we

Table B9: Speech translation results (fine-tuned with 433 hours of data) on LRS3 test.

| PT | PT mod-drop | FT mod | FT mod-drop | AV-BLEU Clean | AV-BLEU Noisy | A-BLEU Clean | A-BLEU Noisy | V-BLEU | Avg-BLEU |
|---|---|---|---|---|---|---|---|---|---|
| *fine-tuned on 433h* | | | | | | | | | |
| ✗ | n/a | AV | ✓ | 10.3 | 4.0 | 9.9 | 2.6 | 2.7 | 5.9 |
| ✓ | ✗ | AV | ✓ | 58.4 | 55.3 | 58.6 | 38.1 | 36.2 | 49.3 |
| ✓ | ✓ | AV | ✓ | 59.4 | 56.2 | 59.2 | 39.0 | 36.1 | 50.0 |
| ✗ | n/a | A | n/a | ✗ | ✗ | 24.5 | 6.8 | ✗ | ✗ |
| ✓ | ✗ | A | n/a | 59.3 | 43.7 | 59.1 | 38.6 | 4.9 | 41.1 |
| ✓ | ✓ | A | n/a | 60.6 | 46.9 | 60.7 | 39.9 | 34.2 | 48.4 |
| ✗ | n/a | V | n/a | ✗ | ✗ | ✗ | ✗ | 1.5 | ✗ |
| ✓ | ✗ | V | n/a | 49.5 | 44.7 | 21.7 | 9.4 | 34.6 | 32.0 |
| ✓ | ✓ | V | n/a | 59.4 | 56.2 | 59.2 | 39.0 | 36.1 | 50.0 |

demonstrate its applications to audio-visual and visual speech recognition in this paper. The former can help hearing-impaired people to better "hear" speech in noisy environments with more accurate transcriptions, while the latter can help people with speech impairment (e.g., aphonia, dysphonia, dysarthria) to "speak" by transcribing silent speech.

For visual speech recognition, there could be concerns about the technology being improperly used for CCTV surveillance. However, current visual speech recognition systems require mostly-frontal and high-resolution videos with a sufficiently high frame rate, such that motions are clearly captured. Hence, the type of data studied for audio-visual speech processing are face-to-face meeting scenarios (AMI, EasyCom, and Ego4D) and recorded speech (LRS3). In contrast, CCTV videos are low resolution, low frame rate, and recorded from angles where faces are mostly not frontal, where visual speech processing models will very likely fail.

# D  More Details of the Baseline ASR, VSR, and AVSR Methods

Table D10 and D11 summarize the model architectures and training/testing setups of the baseline ASR, VSR, and AVSR methods compared in Table 4.

Table D10: Comparison of model architectures with the state-of-the-art ASR/VSR/AVSR models listed in Table 4. T-/S-/(T+S)-{Transformer,Conformer} applies attention over the temporal/spatial/spatial-temporal dimension, respectively. EleAtt-GRU refers to [Zhang et al., 2019].

| Method | Audio Encoder | Video Encoder | Shared Encoder | Decoder |
|---|---|---|---|---|
| Ma et al. [2021], Ma et al. [2022] | 1D-ResNet-18 + T-Conformer | 3D-ResNet-18 + T-Conformer | MLP | T-Conformer |
| Xu et al. [2020] | 1D-CNN + EleAtt-GRU | 3D-CNN + EleAtt-GRU | - | EleAtt-GRU |
| Afouras et al. [2021] | - | 3D-CNN + S-Transformer + T-Transformer | - | T-Transformer |
| Makino et al. [2019] | - | CNN | LSTM | LSTM |
| Serdyuk et al. [2021] | - | (T+S)-Transformer | T-Transformer | LSTM |
| Afouras et al. [2020] | CNN | CNN | - | - |
| [Shi et al., 2022b] [Shi et al., 2022a] u-HuBERT | Linear | 3D-ResNet-18 | T-Transformer | T-Transformer |

Table D11: Comparison of learning paradigms, training objectives, and decoding methods with the state-of-the-art ASR/VSR/AVSR models listed in Table 4

| Method | Type | Training Criterion | Decoding |
|---|---|---|---|
| Ma et al. [2021] | Supervised | LRW pre-training $\rightarrow$ CTC, S2S | CTC+S2S+LM |
| Ma et al. [2022] | Supervised | LRW pre-training $\rightarrow$ CTC, S2S, feature matching from ASR and VSR | CTC+S2S+LM |
| Xu et al. [2020] | Supervised | LRW pre-training $\rightarrow$ S2S, speech enhancement | S2S |
| Afouras et al. [2021] | Supervised | Two-stage curriculum S2S | S2S+LM |
| Makino et al. [2019] | Supervised | RNN-T | RNN-T |
| Serdyuk et al. [2021] | Supervised | RNN-T | RNN-T |
| Afouras et al. [2020] | Semi-Supervised | CTC, knowledge distillation from ASR | CTC+LM |
| [Shi et al., 2022b] [Shi et al., 2022a] u-HuBERT | Self-Supervised | Masked cluster prediction $\rightarrow$ S2S | S2S |