# OpenReview forum: "u-HuBERT: Unified Mixed-Modal Speech Pretraining And Zero-Shot Transfer to Unlabeled Modality"
_NeurIPS.cc/2022/Conference — NeurIPS 2022 Accept_

### Official Review · Reviewer_aZc1 · 2022-07-08

**Rating:** 5
**Confidence:** 3
**Soundness:** 3 good
**Presentation:** 2 fair
**Contribution:** 3 good

**Summary:**

This paper presented unified hidden unit BERT (u-HuBERT), which leverages unlabeled speech data of many different modalities for pre-training, including both unimodal and multimodal speech. The u-HuBERT model is composed of a set of modality-specific feature extractors and a shared transformer backend. With this architecture design, u-HuBERT can learn modality-agnostic frame-level representations, and can be fine-tuned with either multimodal or unimodal data. Experiment results show the effectiveness of u-HuBERT on two speech processing tasks: speech recognition and speaker verification.

-> After reviewing the author rebuttal, I changed my ratings to 5.

**Questions:**

In section 4.3, when showing the results of fine-tuning on OOD labeled LibriSpeech data in Table 3, what's the performance of baseline methods? The author claimed the zero-shot modality transfer performance is decent on LibriSpeech, however, without comparing with baselines, it's hard to convince the readers.

More descriptions need to be added when comparing with SOTA models in section 4.3 (Table 4). Several questions are:
* What are the baseline model settings?
* What are the pre-training/training/fine-tuning approaches used by baseline models?
* What are the datasets?

**Limitations:**

The author had limitation discussion in section 5.

**Strengths And Weaknesses:**

Built upon AV-HuBERT, u-HuBERT can generalize to utilize both multimodal and unimodal data by mapping different inputs into a modality-agnostic feature space using the proposed modality dropout method. Modality dropout also allows zero-shot modality generalization, which means the fine-tuned model can handle modalities in the downstream task that do not cover the ones seen during pre-training. Even though the author showed efficiency of the proposed method in speech recognition and speaker verification tasks, there are still some studies and discussions missing in the paper. For example, the modality dropout rates during pre-training and fine-tuning and what's the effect of different dropout rates in each stage?

Typo
* Table 2 caption "FT-drop-p" should be "FT mod-drop-p".

---

> ### Author Response · Authors · 2022-08-01
> **Response to Reviewer aZc1 (pt. 1)**
>
> We thank the reviewer for the thoughtful and constructive comments. We will incorporate our itemized responses below in the final version of our paper.
>
> **Q1: ​​In section 4.3, when showing the results of fine-tuning on OOD labeled LibriSpeech data in Table 3, what's the performance of baseline methods? The author claimed the zero-shot modality transfer performance is decent on LibriSpeech, however, without comparing with baselines, it's hard to convince the readers.**
>
> We thank the reviewer for the suggestion and have added an additional baseline experiment that fine-tunes a model pre-trained without modality dropout (the same one as “PT=True, PT-mod-drop=False” in Table 2) using LibriSpeech 960h with the identical fine-tuning hyperparameters. The results along with those from Table 3 are presented below.
>
> | PT data | PT mod-drop | FT data | FT mod | AV-Clean-WER | AV-Noisy-WER | A-Clean-WER | A-Noisy-WER | V-WER |
> | --- | --- | --- | --- | --- | --- | --- | --- | --- |
> | LRS3 + VC2-En | False | LS960 | A | 6.2 | 11.0 | 20.9 | 65.9 | 64.7 |
> | LRS3 + VC2-En | True | LS960 | A | 4.5 | 8.9 | 4.8 | 27.6 | 39.4 |
>
> Without pre-training modality dropout, the performances are significantly worse on all five test sets, which is consistent with the trend seen in Table 2 when fine-tuning with audio.

---

> > ### Author Response · Authors · 2022-08-01
> > **Response to Reviewer aZc1 (pt. 2)**
> >
> >
> > **Q2: Even though the author showed efficiency of the proposed method in speech recognition and speaker verification tasks, there are still some studies and discussions missing in the paper. For example, the modality dropout rates during pre-training and fine-tuning and what's the effect of different dropout rates in each stage?**
> >
> > Due to space constraints, we had included details of hyperparameters in Appendix A.2, and had presented less important ablation studies in Appendix B which includes the impact of modality dropout rates during fine-tuning (Table 9) as well as the impact of other factors (learning rate, number of frozen layers, number of steps where the pre-trained module is frozen).
> >
> > We thank the reviewer for suggesting ablation studies on pre-training modality dropout rates. We have conducted additional experiments to study and included the results in the table below. Note that the models were pre-trained with a reduced setup (BASE model pre-trained on 433 hours of LRS3 for 200K updates with 32 GPUs) and fine-tuned with identical configurations using 30 hours of LRS3 without further sweeping the hyperparameters. Hence, the numbers are only comparable with those from the same table, but not directly comparable with the numbers in the paper. In the Table below, “PT AV prob” denotes the probability of the model using both audio and video as input during pre-training, and similarly for “PT A prob” and “PT V prob”.
> >
> > | PT AV prob | PT A prob | PT V prob | FT Setup | AV-Clean-WER | AV-Noisy-WER | A-Clean-WER | A-Noisy-WER | V-WER | Avg-WER |
> > | --- | --- | --- | --- | --- | --- | --- | --- | --- | --- |
> > | 1.00 | 0.00 | 0.00 | AV w/o mod-drop | 5.54 | 14.57 | 17.83 | 67.66 | 59.72 | 33.06 |
> > | 0.70 | 0.15 | 0.15 | AV w/o mod-drop | 5.24 | 15.11 | 6.21 | 47.42 | 50.65 | 24.93 |
> > | 0.50 | 0.25 | 0.25 | AV w/o mod-drop | 5.17 | 15.36 | 6.45 | 48.59 | 51.42 | 25.40 |
> > | 0.50 | 0.15 | 0.35 | AV w/o mod-drop | 6.04 | 17.59 | 7.46 | 51.78 | 51.06 | 26.79 |
> >
> > | PT AV prob | PT A prob | PT V prob | FT Setup | AV-Clean-WER | AV-Noisy-WER | A-Clean-WER | A-Noisy-WER | V-WER | Avg-WER |
> > | --- | --- | --- | --- | --- | --- | --- | --- | --- | --- |
> > | 1.00 | 0.00 | 0.00 | A | 6.26 | 15.47 | 10.67 | 60.81 | 66.89 | 32.02 |
> > | 0.70 | 0.15 | 0.15 | A | 5.83 | 14.23 | 6.10 | 41.29 | 52.13 | 23.92 |
> > | 0.50 | 0.25 | 0.25 | A | 5.50 | 14.84 | 5.96 | 41.72 | 48.72 | 23.35 |
> > | 0.50 | 0.15 | 0.35 | A | 6.22 | 16.00 | 6.54 | 43.71 | 48.71 | 24.24 |
> >
> > | PT AV prob | PT A prob | PT V prob | FT Setup | AV-Clean-WER | AV-Noisy-WER | A-Clean-WER | A-Noisy-WER | V-WER | Avg-WER |
> > | --- | --- | --- | --- | --- | --- | --- | --- | --- | --- |
> > | 1.00 | 0.00 | 0.00 | V | 17.45 | 26.68 | 27.21 | 79.01 | 66.65 | 43.40 |
> > | 0.70 | 0.15 | 0.15 | V | 13.28 | 20.23 | 13.59 | 48.30 | 52.40 | 29.56 |
> > | 0.50 | 0.25 | 0.25 | V | 11.01 | 18.18 | 11.54 | 47.96 | 49.19 | 27.58 |
> > | 0.50 | 0.15 | 0.35 | V | 10.69 | 18.74 | 11.69 | 50.10 | 49.11 | 28.07 |
> >
> > When fine-tuning on audio-visual or audio-only data, among the models pre-trained with modality dropout, setting the probability of V-only higher (PT V prob=0.35) leads to slightly worse performance on AV and A test sets. In contrast, when fine-tuning on visual-only data, setting the probability of V-only higher leads to slightly better average performance. Nevertheless, all three configurations achieve significantly better performance compared to the model pre-trained without modality dropout, confirming that whether modality dropout is used is the key factor to whether zero-shot modality transfer can be successful.
> >
> > Please let us know if the reviewer thinks there are still missing studies or discussions. We will be happy to include them to improve our submission.

---

> > > ### Author Response · Authors · 2022-08-01
> > > **Response to Reviewer aZc1 (pt. 3)**
> > >
> > > **Q3: More descriptions need to be added when comparing with SOTA models in section 4.3 (Table 4). Several questions are: a) What are the baseline model settings? b) What are the pre-training/training/fine-tuning approaches used by baseline models? c) What are the datasets?**
> > >
> > > We have updated the paper and the appendix to include a) what datasets are used for each baseline in Table 4, and b) the model architectures and the training objectives of those baselines in the Appendix D due to space limits.
> > >
> > > As shown in the updated Table 4, most of the state-of-the-art models that report results on LRS3 are supervised and trained on large-scale labeled data including either non-public datasets or filtered datasets that lack sufficient information for reproducing the filtering process. The supervised training pipeline and model architecture also vary significantly among these baselines as indicated in Table 16 and 17 in Appendix D. Hence, following the convention of prior studies, we consider the amount of labeled and unlabeled data used as the key factor when comparing with the literature.
> > >
> > > Furthermore, we would also like to emphasize that Table 4 was presented to highlight the following contributions of this work:
> > > 1. Prior work only studies modality-dependent models. In contrast, u-HuBERT is the first to show state-of-the-art performance on all three modalities with a single model compared to models trained on similar amounts of labeled data.
> > > 2. u-HuBERT is the first model to demonstrate zero-shot modality transfer.
> > > 3. u-HuBERT is the first model to demonstrate unlabeled unimodal data (TD) can benefit not only ASR, but also AVSR and VSR.

---

### Official Review · Reviewer_dtiD · 2022-07-14

**Rating:** 7
**Confidence:** 4
**Soundness:** 3 good
**Presentation:** 3 good
**Contribution:** 3 good

**Summary:**

The paper introduces a self-supevised method in the HuBERT family designed for audio-visual speech, which can be trained on different modalities for pre-training, including single modality audio or visual, and multi-modalities (audio-visual).
u-HuBERT is closely related to AV-HuBERT, but incorporates masked cluster prediction for both unimodal and multimodal data, via masking on modalities, ("modality dropout") which requires the model to predict the same targets using different sets of input modalities.

**Questions:**

I am under the impression that  w2v-BERT produces better representations than HuBERT.  Can you comment on your selection of HuBERT versus w2v-BERT, and whether your proposed additions to AV-HuBERT are applicable to the case of w2v-BERT?

**Ethics Review Area:**

["I don’t know"]

**Limitations:**

Limitations are discussed in section 5

**Strengths And Weaknesses:**



Strengths:

The proposed model provides the ability to fine tune on any modality and work well on all modalities, with comparable performance to modality-specific models.

Evaluation is thorough.
Improvements are small in absolute terms on LRS3, but baseline error rates are very small (e.g., 1.4% WER for AV ASR), so a reduction to 1.2% which is a large relative WER reduction of 14%.  Improvements like this are typically very difficult to achieve.
These are useful results.

The paper is well written and quite clear.


Weaknesses:

The paper is not self-contained, and most of the architectural details are contained in other references.

Novelty relative to AV-HuBERT is not large, but is important.

The proposed methods are practical, but not based on any deep theory developments.

---

> ### Author Response · Authors · 2022-08-01
> **Response to Reviewer dtiD**
>
> We thank the reviewer for the thoughtful and constructive comments. We will incorporate our itemized responses below in the final version of our paper.
>
> **Q1: I am under the impression that w2v-BERT produces better representations than HuBERT. Can you comment on your selection of HuBERT versus w2v-BERT**
>
> We chose HuBERT for two main reasons. First, HuBERT has already been extended to pre-train on audio-visual speech (AV-HuBERT) and benchmarked on ASR, VSR, and AVSR datasets which provides baseline results for comparison. Second, HuBERT is conceptually simpler to train, because it optimizes a single objective (masked cluster prediction), while w2v-BERT is trained by combining three objectives: two from wav2vec2 (contrastive loss and codebook usage) and one from HuBERT (masked cluster prediction).
>
> --------------
>
> **Q2: whether your proposed additions to AV-HuBERT are applicable to the case of w2v-BERT?**
>
> The key to zero-shot modality transfer and pre-training with unimodal data is to learn modality-agnostic features. In u-HuBERT, it is achieved through a combination of “modality dropout” and “masked prediction of the same AV cluster target.” For w2v-BERT, one may achieve this through using audio-visual features as the attractors/distractors for the contrastive loss and quantized audio-visual features as the masked prediction target, while having a copy of the same model with modality dropout to generate the prediction for the contrastive and masked prediction objective.
>
> --------------
>
> **Q3: The paper is not self-contained, and most of the architectural details are contained in other references**
>
> We thank the reviewer for the suggestions and will include more implementation details in the Appendix for completeness.

---

### Official Review · Reviewer_BiUF · 2022-07-17

**Rating:** 3
**Confidence:** 4
**Soundness:** 4 excellent
**Presentation:** 2 fair
**Contribution:** 1 poor

**Summary:**

This paper presents an approach to handle speech signal in multiple modalities within a single unified model that can be pre-trained and fine-tuned with datasets of mixed speech modalities.  The model includes modality specific feature extraction followed by a shared transformer module that learns modality agnostic representations.  Audio and visual modalities are considered and results are presented on speech recognition as well as speaker verification tasks.  It is also shown that the proposed model exhibits strong performance in zero-shot generalization to modalities for which it was not fine-tuned.




**Questions:**

Please see comments above.

**Strengths And Weaknesses:**

This paper is very close to the recently published AV-HuBERT paper [1], so much so that it is unclear to me if the additional contributions are significant enough for publication.  For instance,

* A key novelty of the paper, as mentioned in lines 54-55 and lines 103-104, is that AV-HuBERT pre-trains with audio-visual multimodal data only whereas the proposed approach pre-trains with multimodal as well as unimodal data.  However, as discussed by She et al. [1], a modality dropout is applied during training to make the model learn how to deal with unimodal inputs, thus effectively pre-training on unimodal data as well.

* Lines 121-123 state that “The main idea of this paper is to apply modality dropout to multimodal speech…”.  However modality dropout was applied in [1] as well, and it is unclear how use of modality dropout in current model differs from that in [1].

* Empirical results are also quite close to those reported by Shi et al. [1].  The best ASR results reported in the current paper are AV/V WER of 1.2/27.2% (these include the TD audio-only corpus during pre-training in addition to 1759 hours of audio-visual data) whereas Shi et al. report AV/V WER of 1.3/26.9% where pre-training is done on 1759 hours of audio-visual data.

[1] B. Shi, W. Hsu, K. Lakhotia, A. Mohamed, “Learning audio-visual speech representation by masked multimodal cluster prediction,” ICLR 2022.

The zero-shot generalization results as well as application to speaker-verification task are novel; we don’t have corresponding results from the AV-HuBERT model.

Minor typos:
Line 37: it essential -> it is essential
Lines 54-62: provide reference to AV-HuBERT

---

> ### Author Response · Authors · 2022-08-01
> **Response to Reviewer BiUF (pt. 1)**
>
> We thank the reviewer for the thoughtful and constructive comments. We will incorporate our itemized responses below in the final version of our paper.
>
> **Q1: Empirical results are also quite close to those reported by Shi et al. [1]. The best ASR results reported in the current paper are AV/V WER of 1.2/27.2% (these include the TD audio-only corpus during pre-training in addition to 1759 hours of audio-visual data) whereas Shi et al. report AV/V WER of 1.3/26.9% where pre-training is done on 1759 hours of audio-visual data.**
>
> We would like to note that 1.3% and 26.9% are not the proper numbers to compare to for the following reasons:
> 1. The 26.9% VSR WER is obtained by combining self-supervised learning with *self-training*, where the pre-trained AV-HuBERT model is fine-tuned on 433 hours of labeled LRS3 plus 1,326 hours of ASR-labeled VC2-En. Without self-training, the VSR WER is 28.6% as listed in Table 2 in Shi et al. [2022a]. Moreover, it is the performance of a modality-specific model. For the single-model setup, the same pre-trained model achieves a WER of 29.1% (4th to the last row, Table 4). Under the same single-model setup, our best VSR performance is 27.2% (last row, Table 4). We did not conduct self-training for any results presented in this paper.
> 2. Shi et al. [2022a] did not present AVSR results. The 1.3% WER listed in Shi et al. [2022a] is the ASR result obtained by fine-tuning an Audio HuBERT instead of an AV-HuBERT. Hence, that model can never be fine-tuned for AVSR or VSR. The proper modality-specific AVSR/ASR result of fine-tuning AV-HuBERT is presented in Shi et al. [2022b], which is 1.4%/1.6% as shown in Table 4 in our paper.
>
> In addition, we would also like to kindly refer the reviewer to Reviewer dtiD’s comments
>
> > Improvements are small in absolute terms on LRS3, but baseline error rates are very small (e.g., 1.4% WER for AV ASR), so a reduction to 1.2% which is a large relative WER reduction of 14%. Improvements like this are typically very difficult to achieve.
>
> which points out that even a 0.2% absolute WER reduction is a relatively large WER reduction and difficult to achieve with the baseline error rate of 1.4%.
>
> To confirm the benefit of including additional unlabeled unimodal audio data during pre-training, we have also added additional experiments using LRS3 as unlabeled AV data and VC2-En as unlabeled audio-only data for pre-training. This model and the baseline model pre-trained with only unlabeled audio-visual LRS3 are fine-tuned with 30 hours of labeled audio-visual LRS3 data. The results are shown in the table below, suggesting that adding additional unlabeled audio data brings significant improvements to AVSR.
>
> | PT AV data | PT A data | AV-clean-WER | A-clean-WER | V-WER |
> | --- | --- | --- | --- | --- |
> | LRS3 | - | 6.16 | 6.68 | 44.73 |
> | LRS3 | VC2-En | 5.34 | 6.08 | 43.51 |
>
> --------------
>
> **Q2: A key novelty of the paper, as mentioned in lines 54-55 and lines 103-104, is that AV-HuBERT pre-trains with audio-visual multimodal data only whereas the proposed approach pre-trains with multimodal as well as unimodal data. However, as discussed by She et al. [1], a modality dropout is applied during training to make the model learn how to deal with unimodal inputs, thus effectively pre-training on unimodal data as well. it is unclear how use of modality dropout in current model differs from that in [1].**
>
> We thank the reviewer for pointing out the confusion. In terms of pre-training, the key difference between AV-HuBERT and u-HuBERT is illustrated in Figure 2. The top part of the figure is identical to AV-HuBERT, whereas the bottom part is the extension of u-HuBERT. Specifically, **AV-HuBERT always predicts quantized multimodal features $Y^{av}$** regardless of whether the input is multimodal or simulated unimodal data.
>
> However, if a sample is audio-only speech, one cannot obtain $Y^{av}$. Instead, **only quantized audio features $Y^a$ can be derived from audio-only samples.** Therefore, to incorporate real unimodal speech into pre-training, it is desired that $Y^a$, $Y^v$, and $Y^{av}$ are similar for a given sample, such that there will be less discrepancy between pre-training on simulated unimodal data (which predicts $Y^{av}$) and pre-training on real unimodal data (which predicts $Y^a$ or $Y^v$). This motivates the analysis conducted in Section 4.2, which confirms that pre-training modality dropout leads to producing modal-agnostic features for creating a shared codebook.
>
> Note that AV-HuBERT never predicts $Y^a$ or $Y^v$ because it only pre-trains on multimodal data and all the unimodal inputs are simulated. Therefore, such analysis was not crucial for the setup considered in AV-HuBERT.

---

> > ### Author Response · Authors · 2022-08-01
> > **Response to Reviewer BiUF (pt. 2)**
> >
> > **Q3: This paper is very close to the recently published AV-HuBERT paper [1], so much so that it is unclear to me if the additional contributions are significant enough for publication.**
> >
> > The main contribution of this paper is not proposing new modeling techniques. Instead, **it aims to demonstrate how the existing pre-training modality dropout mechanism enables novel applications that have never been studied before but will have a big impact on the community** as commented by Reviewer yjaa. We summarize the main differences between AV-HuBERT and this paper below:
> >
> > 1. AV-HuBERT [Shi et al., 2022a] is the first SSL model for audio-visual speech achieving state-of-the-art performance on VSR and on ASR as well as AVSR in the follow-up studies [Shi et al., 2022b]. **AV-HuBERT only concerns modality-specific models during fine-tuning, in which modality dropout is not an essential component.** Concretely, in Shi et al. [2022a], modality dropout leads to only 3% relative WER reduction for VSR (57.0% $\rightarrow$ 55.3%) as shown in Table D.1.
> >
> > 2. **u-HuBERT is the first paper to study zero-shot modality transfer for audio-visual speech processing.** For zero-shot modality transfer, we find modality dropout is an essential component for the success, leading to 58% relative WER reduction when fine-tuned on audio-only data (27.8% $\rightarrow$ 11.6%) and 75% relative when fine-tuned on video-only data (47.8% $\rightarrow$ 11.8%) as shown in Table 4. Although AV-HuBERT could achieve it when pre-training modality dropout is activated, it was never studied in Shi et al. [2022a]. **Such innovations are similar to those in GPT-2 [1] and mBART [2], where the models and training objectives are identical to their predecessors, but novel and significant applications are presented.**
> >
> > 3. **u-HuBERT is also the first work to study pre-training with both multimodal and unimodal speech data.** As explained in the answer to Q2 above, we conducted comprehensive analyses comparing the features and codebooks learned by models pre-trained with and without modality dropout. Through understanding the characteristics, we proposed a simple and effective method to incorporate unimodal speech data, which in principle is a superset of the multimodal data where one can obtain in larger quantities.
> >
> > 4. In the Appendix B.1.2, we also conducted extensive ablation studies showing the effectiveness of several fine-tuning techniques to improve the zero-shot modality transfer performance, such as layer freezing.
> >
> > --------------
> >
> > **Q4: The significance of the proposed applications**
> >
> > Regarding the importance of the novel extensions, we would also like to provide a few concrete examples.
> >
> > 1. **Including unlabeled unimodal data for pre-training**: The largest publicly available audio-visual speech datasets are LRS3, AVSpeech, and VoxCeleb, which add up to roughly **4K hours of audio-visual English**. In contrast, there are much more unlabeled audio speech datasets, such as GigaSpeech, People’s Speech, Spotify Dataset, Libri-light, which add up to **>100K hours of audio-only English**. Being able to leverage unlabeled unimodal data is crucial for scaling the model.
> >
> > 2. **Zero-shot modality transfer**: Similarly, there is labeled audio-only data for most of the speech processing tasks, but very little audio-visual data. For example, **there is no labeled audio-visual speech translation data**, but **many available audio-only speech translation datasets** such as CoVoST and Multilingual TEDx The proposed learning paradigm is of great importance to build models that can process audio-visual input without requiring labeled audio-visual data for training.
> >
> > 3. **Single model for many modalities**: There is also a trend of building a single model for different types of data, such as mBART [2], Omnivore [3] in NLP and computer vision communities respectively. We present the first such model for speech and hope to encourage more future research in this direction.
> >
> > [1] Radford, Alec, et al. "Language models are unsupervised multitask learners." OpenAI blog 1.8 (2019): 9.
> >
> > [2] Liu, Yinhan, et al. "Multilingual denoising pre-training for neural machine translation." Transactions of the Association for Computational Linguistics 8 (2020): 726-742.
> >
> > [3] Girdhar, Rohit, et al. "Omnivore: A single model for many visual modalities." Proceedings of the IEEE/CVF Conference on Computer Vision and Pattern Recognition. 2022.

---

> > > ### Author Response · Authors · 2022-08-01
> > > **Response to Reviewer BiUF (pt. 3)**
> > >
> > > **Additional Experiment on Zero-shot Modality Transfer for Speech Translation**
> > >
> > > We conduct additional experiments to demonstrate the effectiveness of zero-shot modality transfer on speech translation. Since there is no publicly available audio-visual speech translation data, we deploy a machine translation model to produce Spanish translation from the English transcript of LRS3. We compare models pre-trained with and without modality dropout used in Table 4, and fine-tune it on 30 hours of LRS3 pseudo-labeled speech translation data with audio and visual input respectively. Each model is evaluated on audio, visual, and audio-visual input, where BLEU scores are reported (the higher, the better).
> > >
> > > | PT data | PT mod-drop | FT data | FT mod | AV-Clean-BLEU | AV-Noisy-BLEU | A-Clean-BLEU | A-Noisy-BLEU | V-BLEU |
> > > | --- | --- | --- | --- | --- | --- | --- | --- | --- |
> > > | LRS3 + VC2-En | False | LRS3-30h | A | 34.27 | 28.52 | 26.85 | 13.74 | 12.07 |
> > > | LRS3 + VC2-En | True | LRS3-30h | A | 40.94 | 28.66 | 41.00 | 23.92 | 24.85 |
> > > | LRS3 + VC2-En | False | LRS3-30h | V | 23.79 | 20.53 | 17.65 | 8.3 | 15.21 |
> > > | LRS3 + VC2-En | True | LRS3-30h | V | 36.63 | 28.09 | 36.07 | 18.24 | 26.10 |
> > >
> > > Results in the table suggests that the proposed technique can also be applied to speech translation.
> > >
> > > --------------
> > >
> > > We hope the reviewer can reconsider the novelty and the significance of this work based on the impact of the proposed applications that were never studied before.

---

> > > > ### Comment · Reviewer_BiUF · 2022-08-07
> > > > **response to authors' comments**
> > > >
> > > > Thanks to the authors for a detailed response and for additional experiments with their proposed u-HuBERT model.  However:
> > > >
> > > > a) Unimodal data during pre-training:  In their response the authors state that “AV-HuBERT always predicts quantized multimodal features.”  But based on Section 3.3 of [1]  “The targets are generated from clustering audio features or features extracted from the previous iteration of the AV-HuBERT model.” Thus, as I understand it, the targets are either audio only (first iteration) or a mix of audio/visual/AV (depending on modality dropout), so while not identical to the proposed unimodal training, the AV-HuBERT paper demonstrated that targets can come from unimodal inputs as well.
> > > >
> > > > b) Zero-shot modality transfer:  Both AV-HuBERT and the proposed u-HuBERT approaches aim to achieve modality agnostic representation in the pre-training step.  So it is expected that both these approaches will achieve good zero-shot modality transfer.  In the related work section authors state that “AV-HuBERT effectively learns modality-specific models (e.g., visual speech recognizer) given labeled data in the target modality for fine-tuning. In contrast, this work focuses on building a single model for all modalities seen during pre-training with labeled data in any subset of modalities.”  However, when the AV-HuBERT model is fine-tuned on one modality, what is its performance on other modalities, and how well does AV-HuBERT work as a ‘single model for all modalities’?  The zero-shot modality transfer performance of AV-HuBERT is a key missing baseline.
> > > >
> > > > Overall, I still think that the significance of additional contributions is not enough for publication and would like to maintain my original rating.
> > > >
> > > > [1] B. Shi, W. Hsu, K. Lakhotia, A. Mohamed, “Learning audio-visual speech representation by masked multimodal cluster prediction,” ICLR 2022.

---

> > > > > ### Author Response · Authors · 2022-08-08
> > > > > **Response to Reviewer BiUF's Followup Comments**
> > > > >
> > > > > We thank the reviewer for engaging in the discussion. Our itemized responses are included below
> > > > >
> > > > > > Q1 **Unimodal data during pre-training**: Thus, as I understand it, the targets [of AV-HuBERT] are either audio only (first iteration) or a mix of audio/visual/AV (depending on modality dropout) … the AV-HuBERT paper demonstrated that targets can come from unimodal inputs as well.
> > > > >
> > > > > This is incorrect. [1] always uses AV targets after the first iteration. As put in the fifth paragraph in Section 3.3 titled “Audio-visual clustering” and Section E.1 in [1], the goal of pre-training a AV-HuBERT is to generate multimodal cluster as targets, because “Combining both modalities produces cluster assignments of higher quality for
> > > > > AV-HuBERT.” This is corroborated by inspecting the open-sourced code (https://github.com/facebookresearch/av_hubert/blob/main/avhubert/clustering/dump_hubert_feature.py).
> > > > >
> > > > > Thus, [1] has never demonstrated that targets can come from unimodal input. The reasons why they did not have to demonstrate that is because multimodal cluster assignments are claimed to be better and [1] only pre-trains on multimodal data. For completeness, the reason why [1] uses audio target in the first iteration is because high quality audio-visual feature extractor (AV-HuBERT) is only available after the first iteration.
> > > > >
> > > > >
> > > > > > Q2 **Zero-shot modality transfer**: However, when the AV-HuBERT model is fine-tuned on one modality, what is its performance on other modalities, and how well does AV-HuBERT work as a ‘single model for all modalities’? The zero-shot modality transfer performance of AV-HuBERT is a key missing baseline.
> > > > >
> > > > > u-HuBERT is a generalization of AV-HuBERT in terms of what data can be used for pre-training. When it is trained on only audio-visual speech, it is identical to AV-HuBERT.
> > > > >
> > > > > On the other hand, zero-shot modality transfer is a novel pre-training/fine-tuning/testing paradigm we proposed which is enabled when pre-training modality dropout is applied. In other words, it works for both u-HuBERT and a special case of u-HuBERT, AV-HuBERT, where only multimodal data is used.
> > > > >
> > > > > Regarding the results, the u-HuBERT from the fourth-to-the-last row to the second-to-the-last row are pre-trained without unimodal data, and are effectively AV-HuBERT. In the last row, we demonstrated that adding additional unimodal data leads to improvement on all three modalities.
> > > > >
> > > > > To conclude, we did not claim “only u-HuBERT can achieve zero-shot modality transfer but not AV-HuBERT.” Instead, what we claimed and empirically verified are __pre-training dropout enables zero-shot transfer (for both u-HuBERT and its special case AV-HuBERT)__.
> > > > >
> > > > > This pre-training/fine-tuning/testing paradigm which enables training an audio-visual speech recognition model using only labeled audio data is not explored or even discussed in [1], even though it can be achieved with AV-HuBERT. We would argue that this is similar to the innovation of the zero-shot / few-shot learning ability in large language models where novel applications are derived from existing models.
> > > > >
> > > > >
> > > > > > Q3: **Modality-specific vs single model**: In the related work section authors state that “AV-HuBERT effectively learns modality-specific models (e.g., visual speech recognizer) given labeled data in the target modality for fine-tuning. In contrast, this work focuses on building a single model for all modalities seen during pre-training with labeled data in any subset of modalities.”
> > > > >
> > > > > The preceding sentence of the quoted ones above is “The fine-tuning and evaluation protocols also differ significantly.” We meant to emphasize the difference in terms of __the evaluation protocols and the applications considered__ between the two papers ([1] versus this paper), not the two models (AV-HuBERT versus u-HuBERT). We apologize for the confusion and would correct that in the subsequent version.
> > > > >
> > > > > -------------------
> > > > >
> > > > > We are happy to discuss with the reviewer regarding any additional concerns or ambiguities comparing the two papers. Thank you for your consideration

---

### Official Review · Reviewer_yjaa · 2022-07-23

**Rating:** 7
**Confidence:** 4
**Soundness:** 4 excellent
**Presentation:** 4 excellent
**Contribution:** 4 excellent

**Summary:**

This paper presents a self-supervised learning framework, termed u-HuBERT, that can utilize both multimodal and unimodal unsupervised speech data including audio-visual, audio and visual to pre-train a single model by utilizing modality dropout. If this pre-trained model is fine-tuned on audio-visual data with modality dropout, it can achieve comparable or better results than the state-of-the-art modality-specific models. In addition, the authors show that the modality dropout used in pre-taining is important for learning modality agnostic features, which makes the model only fine-tuned on audio perform well on audio-visual or visual speech data, achieving zero-shot modality generation for speech recognition and speaker verification.

**Questions:**

Questions/suggestions:
- The proposed framework assumes the multiple modalities features align in a frame-by-frame fashion. How can it be applied to the case where the assumption does not hold, e.g. between the audio and text modalities?
- Line 194: Is the pre-training modality dropout probability 0.25/0.25 tuned? How sensitive is its value to the results?
- In Table 2, modality dropout on fine-tuning also improves AV-WER for the models fine-tuned on AV data. It would be better to explain why that is the case.
- Line 259: For the claim of u-HuBERT "achieving on par or better than the best modality-specific models", it would be better to mention the caveat as well: it's slightly worse than the best modality-specific visual speech WER (27.2 vs. 25.9) although using much less labeled visual speech training data.

Typos:
- Line 37: it essential -> it is essential
- Line 327: supported u-HuBERT -> supported by u-HuBERT

**Limitations:**

Yes, the authors have addressed the limitations in Section 5 and the potential negative societal impact of their work in Appendix C.

For completeness, the authors should also mention another limitation of the system that assumes all the modalities align in a frame-by-frame fashion (as mentioned in my question above), or discuss possible extensions to address this limitation for cases where this assumption does not hold, e.g. between audio and text modalities.

**Strengths And Weaknesses:**

Originality:

- Previous work on audio-visual ASR are mainly focusing on modality-specific training / fine-tuning. This work, using a shared target token space for all unimodal and multimodal data in a single model, and find-tuning it with labeled data only on a subset of the modalities seen during pre-training but perform well on other modalities in test to achieve zero-shot modality generalization, is a novel idea.

- Even though modality dropout has been used in the literature, the analysis to show that it is important for learning modality agnostic features and achieving huge gains in zero-shot modality generalization scenarios is new.

Quality:

- The authors show that the proposed single model achieves on par or better results than state-of-the-art modality-specific models for speech recognition. It is very promising that the model fine-tuned only on audio data performs well on audio-visual and visual speech data with comparable results to the models fine-tuned on the modality-specific data.

- The experiment using out-of-domain labeled audio data from LibriSpeech for fine-tuning shows convincing results to support the claim that the model's zero-shot modality transfer is not achieved through memorization.

- The proposed model also shows the zero-shot modality generalization ability but only to some extent. The authors suggest this is due to the nature of the speaker verification task, where each modality can only infer a subset of the speaker features. This demonstrates a limitation to the zero-shot transfer.

Clarity:

- The paper is written clearly. The presentation is very well organized.

Significance:

- This work contributes to the community significantly, by showing an approach to pre-train on both multimodal and unimodal speech data to not only improve the state-of-the-art results in audio-visual ASR but also achieve high-quality zero-shot modality generalization. In particular, it brings significant value to the tasks with modalities without much labeled data.

---

> ### Author Response · Authors · 2022-08-01
> **Response to Reviewer yjaa (pt. 1)**
>
> We thank the reviewer for the thoughtful and constructive comments. We will incorporate our itemized responses below in the final version of our paper.
>
> **Q1: The proposed framework assumes the multiple modalities features align in a frame-by-frame fashion. How can it be applied to the case where the assumption does not hold, e.g. between the audio and text modalities?**
>
> We believe the idea can be easily extended to pairs of modalities that are **temporally monotonically aligned**, such as audio and its text transcript, with the help of an alignment module [1] and a duration prediction module [2] to create frame-level text representations.
>
> Concretely, when pre-training on multimodal data (paired speech and text), the alignment module can align each phone or word in the text transcript to the corresponding speech frame; hence speech and text features can be frame-by-frame fused. When pre-training on unimodal text data, the text sequence can be transformed into a frame-level sequence to match the length of speech sequences by repeating each phone or word based on the predicted duration.
>
> [1] McAuliffe, Michael, et al. "Montreal Forced Aligner: Trainable Text-Speech Alignment Using Kaldi." Interspeech. Vol. 2017. 2017.
>
> [2] Ren, Yi, et al. "Fastspeech 2: Fast and high-quality end-to-end text to speech." arXiv preprint arXiv:2006.04558 (2020).
>
> --------------
>
> **Q2: Is the pre-training modality dropout probability 0.25/0.25 tuned? How sensitive is its value to the results?**
>
> We followed the pre-training modality dropout configuration used in Shi et al. [2022a] and did not further tune it. Additional results studying the sensitivity of pre-training modality dropout configurations are listed below. Note that the models were pre-trained with a reduced setup (BASE model pre-trained on 433 hours of LRS3 for 200K updates with 32 GPUs) and fine-tuned with identical configurations using 30 hours of LRS3 without further sweeping the hyperparameters. Hence, the numbers are only comparable with those from the same table, but not directly comparable with the numbers in the paper. In the Table below, “PT AV prob” denotes the probability of the model using both audio and video as input during pre-training, and similarly for “PT A prob” and “PT V prob”.
>
> | PT AV prob | PT A prob | PT V prob | FT Setup | AV-Clean-WER | AV-Noisy-WER | A-Clean-WER | A-Noisy-WER | V-WER | Avg-WER |
> | --- | --- | --- | --- | --- | --- | --- | --- | --- | --- |
> | 1.00 | 0.00 | 0.00 | AV w/o mod-drop | 5.54 | 14.57 | 17.83 | 67.66 | 59.72 | 33.06 |
> | 0.70 | 0.15 | 0.15 | AV w/o mod-drop | 5.24 | 15.11 | 6.21 | 47.42 | 50.65 | 24.93 |
> | 0.50 | 0.25 | 0.25 | AV w/o mod-drop | 5.17 | 15.36 | 6.45 | 48.59 | 51.42 | 25.40 |
> | 0.50 | 0.15 | 0.35 | AV w/o mod-drop | 6.04 | 17.59 | 7.46 | 51.78 | 51.06 | 26.79 |
>
> | PT AV prob | PT A prob | PT V prob | FT Setup | AV-Clean-WER | AV-Noisy-WER | A-Clean-WER | A-Noisy-WER | V-WER | Avg-WER |
> | --- | --- | --- | --- | --- | --- | --- | --- | --- | --- |
> | 1.00 | 0.00 | 0.00 | A | 6.26 | 15.47 | 10.67 | 60.81 | 66.89 | 32.02 |
> | 0.70 | 0.15 | 0.15 | A | 5.83 | 14.23 | 6.10 | 41.29 | 52.13 | 23.92 |
> | 0.50 | 0.25 | 0.25 | A | 5.50 | 14.84 | 5.96 | 41.72 | 48.72 | 23.35 |
> | 0.50 | 0.15 | 0.35 | A | 6.22 | 16.00 | 6.54 | 43.71 | 48.71 | 24.24 |
>
> | PT AV prob | PT A prob | PT V prob | FT Setup | AV-Clean-WER | AV-Noisy-WER | A-Clean-WER | A-Noisy-WER | V-WER | Avg-WER |
> | --- | --- | --- | --- | --- | --- | --- | --- | --- | --- |
> | 1.00 | 0.00 | 0.00 | V | 17.45 | 26.68 | 27.21 | 79.01 | 66.65 | 43.40 |
> | 0.70 | 0.15 | 0.15 | V | 13.28 | 20.23 | 13.59 | 48.30 | 52.40 | 29.56 |
> | 0.50 | 0.25 | 0.25 | V | 11.01 | 18.18 | 11.54 | 47.96 | 49.19 | 27.58 |
> | 0.50 | 0.15 | 0.35 | V | 10.69 | 18.74 | 11.69 | 50.10 | 49.11 | 28.07 |
>
> When fine-tuning on audio-visual or audio-only data, among the models pre-trained with modality dropout, setting the probability of V-only higher (PT V prob=0.35) leads to slightly worse performance on AV and A test sets. In contrast, when fine-tuning on visual-only data, setting the probability of V-only higher leads to slightly better average performance. Nevertheless, all three configurations achieve significantly better performance compared to the model pre-trained without modality dropout, confirming that whether modality dropout is used is the key factor to whether zero-shot modality transfer can be successful.

---

> > ### Author Response · Authors · 2022-08-01
> > **Response to Reviewer yjaa (pt. 2)**
> >
> >
> > **Q3: In Table 2, modality dropout on fine-tuning also improves AV-WER for the models fine-tuned on AV data. It would be better to explain why that is the case**
> >
> > The relevant results from Table 2 are summarized in the Table below:
> >
> > | Pre-training Setup | AV-WER on Clean (w/o $\rightarrow$ w/ FT mod-drop) | AV-WER on Noisy (w/o $\rightarrow$ w/ FT mod-drop) |
> > | ---  | ---  | ---  |
> > | No pre-training | 3.8 $\rightarrow$ 3.6  | 17.2 $\rightarrow$ 15.9 |
> > | Pre-training w/ mod-drop | 1.3 $\rightarrow$ 1.3  | 4.8 $\rightarrow$ 4.1 |
> > | Pre-training w/o mod-drop | 1.2 $\rightarrow$ 1.3  | 5.2 $\rightarrow$ 4.6 |
> >
> > On the clean test set, fine-tuning modality dropout leads to 0.2% absolute WER reduction if the weights are not initialized from a pre-trained model, but leads to 0.1% absolute WER increase for the model pre-trained with modality dropout. In contrast, on the noisy test set, fine-tuning modality dropout results in consistent and significant improvements for all three pre-training setups (1.3%, 0.7%, 0.6% absolute WER reduction, respectively).
> >
> > The significant improvement in noisy conditions can be attributed to the fact that in those conditions, the model would need to rely more on the visual input, and with fine-tuning modality dropout, the model is encouraged to utilize visual information more since the audio stream is sometimes dropped. On the other hand, the minor improvements on the clean set for the “No pre-training” model could result from using modality dropout as a data augmentation method for regularizing a large model that is not pre-trained.
> >
> > --------------
> >
> > **Q4: For the claim of u-HuBERT "achieving on par or better than the best modality-specific models", it would be better to mention the caveat as well: it's slightly worse than the best modality-specific visual speech WER (27.2 vs. 25.9) although using much less labeled visual speech training data.**
> >
> > We thank the reviewer for the advice and will clarify it.

---

### Comment · Area_Chair_AnbA · 2022-08-08
**Reviewers yjaa, dtiD, and aZc1  - please reply to the authors**

The authors have responded to questions that you raised:
- yjaa asked about tasks where framewise alignment between modalities is not present, whether the modality dropout is tuned and how sensitive results are to this hyperparameter, why modality dropout improves AV-WER, and cautions the authors against overclaiming
- dtiD asks why HuBERT was selected for the study instead of w2v-BERT, whether or not the additions to AV-HuBERT would also apply to w2v-BERT, and points out that the paper is not sufficiently self contained
- aZc1 asks about the performance of the Librispeech baselines, points out that a lot of questions about the influence of modality dropout rates are left unaddressed, and requests clarifications in the comparison to SOTA methods

As we are near the end of the author-reviewer discussion period, I ask that you please read the authors' reply and respond promptly.

Thank you to reviewer BiUF for already having responded to the authors

---

### Meta-Review · Area_Chair_AnbA · 2022-08-26

**Recommendation:** Accept
**Confidence:** Less certain

**Metareview:**

This paper enjoyed a reasonable discussion between the authors and the reviewers, and the authors have improved the paper in a number of areas, including (1) more experiments clarifying the role of the modality dropout rates on model performance, (2) a baseline for the Librispeech transfer experiment, and (3) more details in the comparisons to other state-of-the-art models.

While the reviewers generally agree that the zero-shot results are novel, the experimental results are strong, and the analysis of the role of modality dropout in learning modality-agnostic representations is a good contribution, they split on their final recommendation on the paper, with three recommending acceptance (at least to some degree) and one sticking to a borderline reject recommendation after discussion with the authors, arguing that the novelty with respect to the AV-HuBERT paper is insufficient.

I am recommending that the paper be accepted, but I think the discussion between the authors and reviewer BiuF includes important points that should be made more strongly in the paper. Specifically, they need to emphasize more strongly the use of targets from multiple modalities (their answer to Q1 in "Response to Reviewer BiUF's Followup Comments"), and they need to explain *in the paper* that "the u-HuBERT from the fourth-to-the-last row to the second-to-the-last row are pre-trained without unimodal data, and are effectively AV-HuBERT" (from their answer to Q2 in "Response to Reviewer BiUF's Followup Comments") to help readers understand the relationship between u-HuBERT and AV-HuBERT. I would also suggest that they add experiments where modality dropout is applied during cluster label generation as an additional constrast.


**Award:**

No

---

### Decision · Program_Chairs · 2022-09-14

Accept